# Single cell T cell landscape and T cell receptor repertoire profiling of AML in context of PD-1 blockade therapy

Hussein A. Abbas [1,2,13], Dapeng Hao [1,3,13], Katarzyna Tomczak[3,13], Praveen Barrodia [3], Jin Seon Im[4,5], Patrick K. Reville[1], Zoe Alaniz [2], Wei Wang[6], Ruiping Wang[3], Feng Wang[3], Gheath Al-Atrash[4,5], Koichi Takahashi [2,3], Jing Ning[7], Maomao Ding[7,8], Hannah C. Beird [3], Jairo T. Mathews[2], Latasha Little[3], Jianhua Zhang [3], Sreyashi Basu [9], Marina Konopleva [2], Mario L. Marques-Piubelli [10], Luisa M. Solis [10], Edwin Roger Parra [10], Wei Lu[10], Auriole Tamegnon[10], Guillermo Garcia-Manero [2], Michael R. Green [3,11], Padmanee Sharma [9,12], James P. Allison[9], Steven M. Kornblau [2], Kunal Rai [3✉], Linghua Wang [3,5✉], Naval Daver[2✉] & Andrew Futreal [3,5✉]

In contrast to the curative effect of allogenic stem cell transplantation in acute myeloid leukemia via T cell activity, only modest responses are achieved with checkpoint-blockade therapy, which might be explained by T cell phenotypes and T cell receptor (TCR) repertoires. Here, we show by paired single-cell RNA analysis and TCR repertoire profiling of bone marrow cells in relapsed/refractory acute myeloid leukemia patients pre/post azacytidine +nivolumab treatment that the disease-related T cell subsets are highly heterogeneous, and their abundance changes following PD-1 blockade-based treatment. TCR repertoires expand and primarily emerge from CD8$^+$ cells in patients responding to treatment or having a stable disease, while TCR repertoires contract in therapy-resistant patients. Trajectory analysis reveals a continuum of CD8$^+$ T cell phenotypes, characterized by differential expression of granzyme B and a bone marrow-residing memory CD8$^+$ T cell subset, in which a population with stem-like properties expressing granzyme K is enriched in responders. Chromosome 7/ 7q loss, on the other hand, is a cancer-intrinsic genomic marker of PD-1 blockade resistance in AML. In summary, our study reveals that adaptive T cell plasticity and genomic alterations determine responses to PD-1 blockade in acute myeloid leukemia.

[1] Division of Cancer Medicine, Medical Oncology Fellowship, University of Texas M D Anderson Cancer Center, Houston, TX, USA. [2] Department of Leukemia, University of Texas MD Anderson Cancer Center, Houston, TX, USA. [3] Department of Genomic Medicine, University of Texas M D Anderson Cancer Center, Houston, TX, USA. [4] Department of Stem Cell Transplantation and Cellular Therapy, University of Texas MD Anderson Cancer Center, Houston, TX, USA. [5] Graduate School of Biomedical Sciences, The University of Texas MD Anderson Cancer Center, Houston, TX, USA. [6] Department of Hematopathology, University of Texas MD Anderson Cancer Center, Houston, TX, USA. [7] Department of Biostatistics, University of Texas MD Anderson Cancer Center, Houston, TX, USA. [8] Department of Statistics, Rice University, Houston, TX, USA. [9] Department of Immunology, University of Texas MD Anderson Cancer Center, Houston, TX, USA. [10] Department Translational Molecular Pathology, University of Texas MD Anderson Cancer Center, Houston, TX, USA. [11] Department of Lymphoma and Myeloma, University of Texas MD Anderson Cancer Center, Houston, TX, USA. [12] Department of Genitourinary Medical Oncology, University of Texas MD Anderson Cancer Center, Houston, TX, USA. [13] These authors contributed equally: Hussein A. Abbas, Dapeng Hao, Katarzyna Tomczak. ✉email: KRai@mdanderson.org; lwang22@mdanderson.org; NDaver@mdanderson.org; AFutreal@mdanderson.org

Despite originating in an immune rich bone marrow (BM) environment, AML cells disrupt normal hematopoiesis and evade immune surveillance[1,2]. However, leukemic cells are susceptible to immune-mediated eradication. Specifically, alloSCT remains the only curative option for patients with AML, largely achieved via the grafted T cells versus leukemia effect[3]. AlloSCT is not a viable option for many AML patients who have comorbidities or lack matched donors. Leveraging the patient's own immune T cells is an attractive alternative for augmenting AML therapeutic strategies.

Immune checkpoint inhibition therapy have transformed outcomes of solid cancer patients[4–7]. Judging by the efficacy of alloSCT, AML would be hypothesized to be an ideal immune responsive tumor. Yet, CTLA4-blockade with ipilimumab following alloSCT[8], and combining the hypomethylating agent azacitidine with the PD-1 inhibitor nivolumab (hereafter referred to as ICB-based therapy) in R/R AML[9] demonstrated modest and variable efficacies in patients. This underscores a compelling need to decipher the T cell landscape of AML in the context of PD-1 blockade therapy, similar to previous work in solid cancers[10–13].

Single cell RNA (scRNA) profiling is a powerful tool to guide our interpretation of cellular diversity and T cell states[1,11–14]. Applying scRNA to dissect the solid cancers landscape following immune checkpoint therapy informed cellular mechanisms of therapeutic response. For instance, melanoma resident CD8+ TCF7+ cells, abundance of dysfunctional CD8 + T cells, and accumulation of exhausted T cells in melanoma correlated with improved responses to immunotherapies[15–18]. However, T cell functionality is effective if the intratumoral T cell receptor (TCR) repertoire is intrinsically tumor reactive[19]. Thus, paired analysis with scRNA and scTCR adds an orthogonal dimension to further characterize T cell states and responses to therapy. In solid cancers, TCR repertoire profiling is used to examine the intratumoral T cell responses and as a biomarker of response to immune checkpoint therapy[19–28], and revealed the expansion of novel clones in skin cancers following PD-1 blockade[28]. While generalizing these observations is tempered by small patient numbers, the depth of the analysis allows for high resolution dissection of T cells and their repertoires. In AML and other hematologic malignancies, the dynamics of TCR repertoires in context of immune checkpoint blockade therapy are largely unexplored. It is yet to be demonstrated that degree of responses to immune therapies in AML are indeed related to an adaptive T cell repertoire.

Here, we show that responses to ICB-based therapy in R/R AML are associated with expansions and novel emergence of CD8 + T-cell clonotypes while resistance to this therapy is associated with contracted T-cell clonotypes. Further, CD8 + T cells in AML are on continuum with GZMK expression being enriched in a memory subset of CD8 + T cells. Our work reveals that deletion in chromosome 7/7q is an intrinsic AML biomarker of resistance to ICB-therapy. Findings from this analysis afford a deep characterization of AML T-cell landscape and can be extrapolated to interpret the T cell dynamics in response to PD-1 blockade-therapy in other hematologic malignancies by identifying BM residing T cell subsets and tumor intrinsic factors that could be leveraged therapeutically.

## Results

**Patient cohort and characteristics.** We conducted paired scRNA and scTCR profiling on 22 (8 pre- and 14 post-treatment) BM aspirates from 8 R/R AML treated with ICB-based therapy on NCT02397720 (Fig. 1A). Clinical and demographic characteristics are shown in Supplementary Fig. 1A and Supplementary Table 1. Briefly, prior to receiving ICB-based therapy, 7 of 8

patients progressed on hypomethylating agents. While on ICB-based treatment, 3/8 patients (PT1-3) responded, while 3/8 (PT4-6) were non-responders (NR) and 2/8 patients had stable disease (SD) (Fig. 1A). Six out of 8 patients had at least one cytogenetic abnormality prior to ICB-based therapy initiation, including 3/3 patients (non-responders) harboring chromosome 7/7q (chr7/7q) deletion (Supplementary Table 1). Targeted DNA sequencing in at least 1 timepoint per patient (total evaluated 17/22) revealed mutations in *ASXL1* (4/8 patients), *TET2* (3/8 patients), *SRSF2* (3/8 patients) and *FLT3* (2/8 patients) (Supplementary Fig. 1B).

**Cluster definitions in healthy and AML BMs.** To guide cluster annotation in AML BMs, we generated a BM cell reference from 13,633 cells from 2 healthy BM donors and utilized canonical immune gene markers[29–33], as previously done[28,34–36] (Fig. 1B). UMAP and trajectory analysis[37–39] demonstrated a differentiation spectrum originating from hematopoietic stem/progenitor cells (Fig. 1C and Supplementary Fig. 2A), consistent with previous reports[29,40]. To further refine our analysis, we then mapped healthy BM cells to an independent reference dataset of 30,672 healthy BM cells[41] and found 91% concordance for cell annotations (Supplementary Fig. 2B–E), confirming accurate cluster definitions.

To distinguish AML cells from other cellular constituents of the BM environment, we removed cells that are doublets, have low read-depth, or high mitochondrial gene expression as previously recommended[29,36,42,43], then applied canonical markers to define AML clusters. We further verified our quality control measures of doublets by applying DoubletFinder[44]. Consistent with our quality control analysis, the doublet score distribution did not demonstrate doublet clustering, suggesting that doublet cells were appropriately removed in our initial quality control analysis (Supplementary Fig. 2F, G). We used expression pattern of multiparametric flow cytometry and immunohistochemistry markers of same timepoint, when available, for confirming AML clusters (Supplementary Fig. 3A–D). To further refine our approach, we computationally combined each AML BM with healthy donor BMs, and demonstrated distinct cluster formation for AML cells away from healthy cells (Fig. 1D). We further validated putative AML cells by inferring aneuploidy status using *inferCNV* tool[45] from scRNAseq and demonstrated concordance with clinical cytogenetics profile (Supplementary Table 1) (Fig. 1E).

A total of 60,753 AML and 52,641 tumor microenvironment (TME) cells from the 22 R/R AML BM aspirates of 8 patients passed quality assessment and were included in downstream analysis. The AML cells proportion identified using scRNAseq closely correlated with blast proportion measured via clinical flow cytometry ($r = 0.87$, $p = 1.5 \times 10^{-7}$) and histopathology ($r = 0.73$, $p = 0.0001$) (Fig.1F). Pre- and post-treatment AML cells clustered by patient (Fig. 1G), while TME components from different patients clustered together and had different distributions (Fig. 1H, I). The clustering patterns of AML and TME cells were similar to other cancers displaying intertumoral heterogeneity[28,29,35,46,47].

**Variable capacity for TCR clonotype expansions following treatment.** TCR profiling can reflect T cell activities in response to checkpoint-blockade therapies[19–28]. We performed α and β scTCR profiling in 26,095 T cells from 22 BM aspirates of 8 AML patients before and after ICB-based treatment, and in 4742 T cells from healthy BMs. Only 7.2% (345/4742) of TCR clonotypes in healthy BMs were shared in 2 or more T cells, compared to 51% of TCR clonotypes in all AML BMs. The clonotype size, represented by number of cells expressing the same TCR sequence, ranged from 1 to 16 in healthy donors, compared to 1 to 1200 in

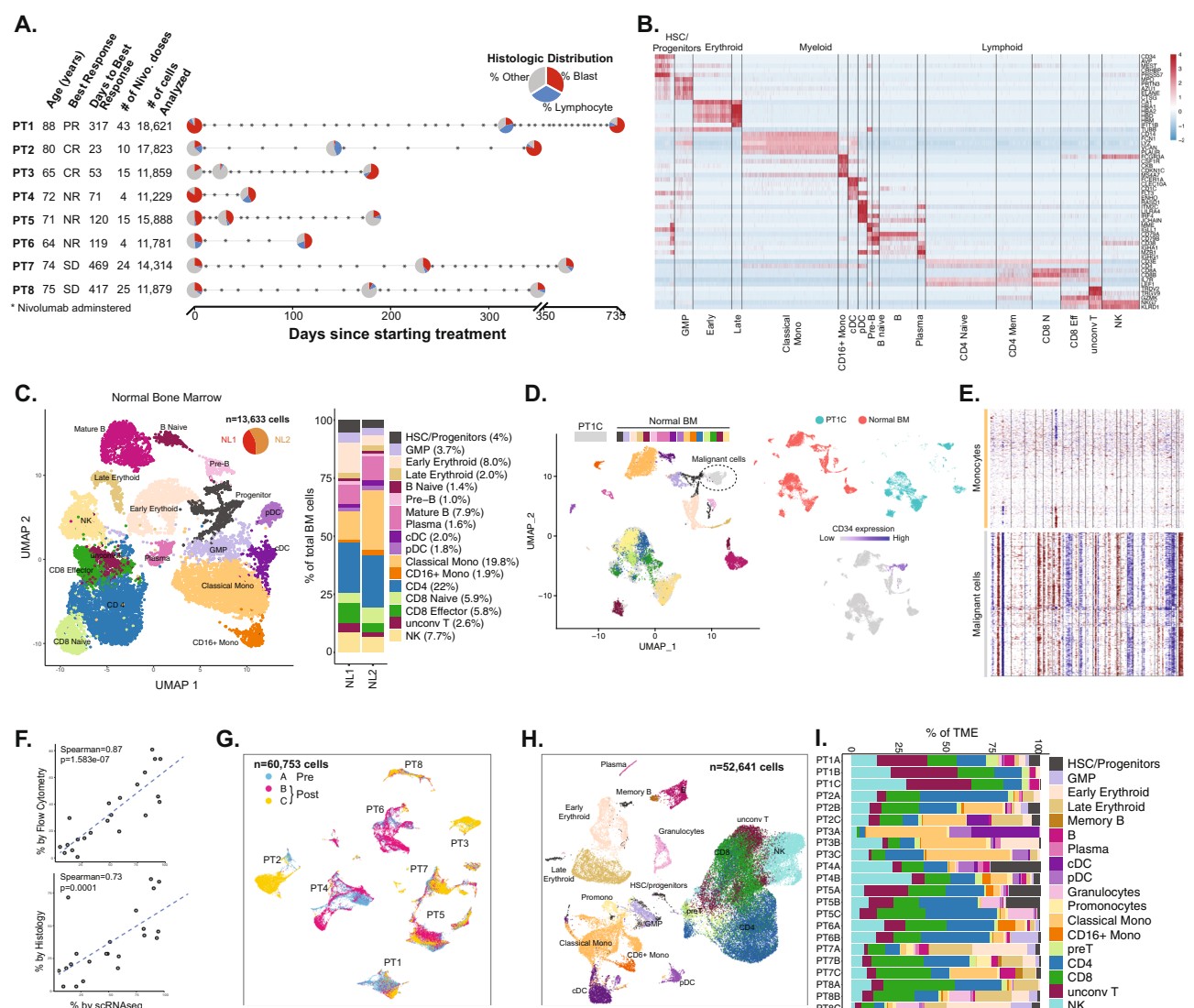

**Fig. 1 Study design and single cell assessment of AML and healthy bone marrows. A** Clinical design summarizing the age, response type per ELN, time to response, treatment frequency and the number of cells analyzed per patient. **B** Canonical gene expression markers to define the healthy bone marrow (BM) and tumor microenvironment (TME) cellular subsets. **C** UMAP-based analysis of healthy BM cellular components with frequency of each cell type. **D** Representative output of combining patient BM sample (represented by PT1, post-treatment timepoint C) with healthy BM donor cells demonstrating distinct clustering of tumor cells with concordant expression of CD34 confirmed by flow cytometry and immunohistochemistry, when available (Supplementary Fig. 3A–D). **E** Representative Inferring copy number variation of malignant cells compared to monocytes demonstrate concordant cytogenetic profiling, further confirming AML cell identity. **F** Spearman correlation between the number of cells detected by scRNA versus flow cytometry and histopathology. **G** UMAP clustering of AML cells. **H** UMAP clustering of TME components. **I** Distribution of TME components in AML patients at different timepoints (A is pre-treatment, B and C are post-treatment samples). *PR* partial response, *CR* complete response, *NR* no response, *SD* stable disease, *HSC* hematopoietic stem cell, *GMP* granulocyte–monocytic progenitor, *cDC* conventional dendritic cell, *pDC* plasmacytoid dendritic cell, *unconv T* unconventional T, *NK* natural killer.

AML (Fig. 2A), indicating higher clonality in AML bone marrows.

At pretreatment timepoints, there was marked variation in frequency of T cells contributing to the most abundant clonotype in AML patients. Following treatment, 4 patients (2 responders and 2 SD) had expansion of their most abundant clonotypes (Fig. 2B). Conversely, NR (3/3) patients had contraction of their most abundant clonotypes (Fig. 2B). As expected, healthy BM T-cell repertoire had few dominant clones (Fig. 2B). Compared to pre-treatment (timepoint A), the 3 responders had an increase in their clonality following treatment (timepoint B) that persisted (timepoint C), except for PT3 (responder) who initially had an increase in clonality (timepoint B), followed by a subsequent decrease in clonality (timepoint C) prior to morphologic AML

progression. Three NR and 1 SD patients had relative reduction in their TCR clonality following treatment, while 1 SD patient (PT7) had a persistent increase in TCR clonality (Fig. 2C).

T cells are adaptable and the capacity for TCR repertoire treatment reactivity is variable[19]. To investigate how the clonotype abundance changed following treatment, we compared frequencies of each clone and identified significantly changed ones following treatment (Fisher's exact $p < 0.05$) (Fig. 2D). Novel clones (not detected at pre-treatment) constituted 43% of significantly expanded clonotypes (Fisher's exact $p < 0.05$) (Fig. 2D, E). Association with clinical response revealed different patterns of clonotype change in abundance. Specifically, the majority (77%) of significantly expanded clones were found in responders (77%) and patients with SD (18%), whereas the

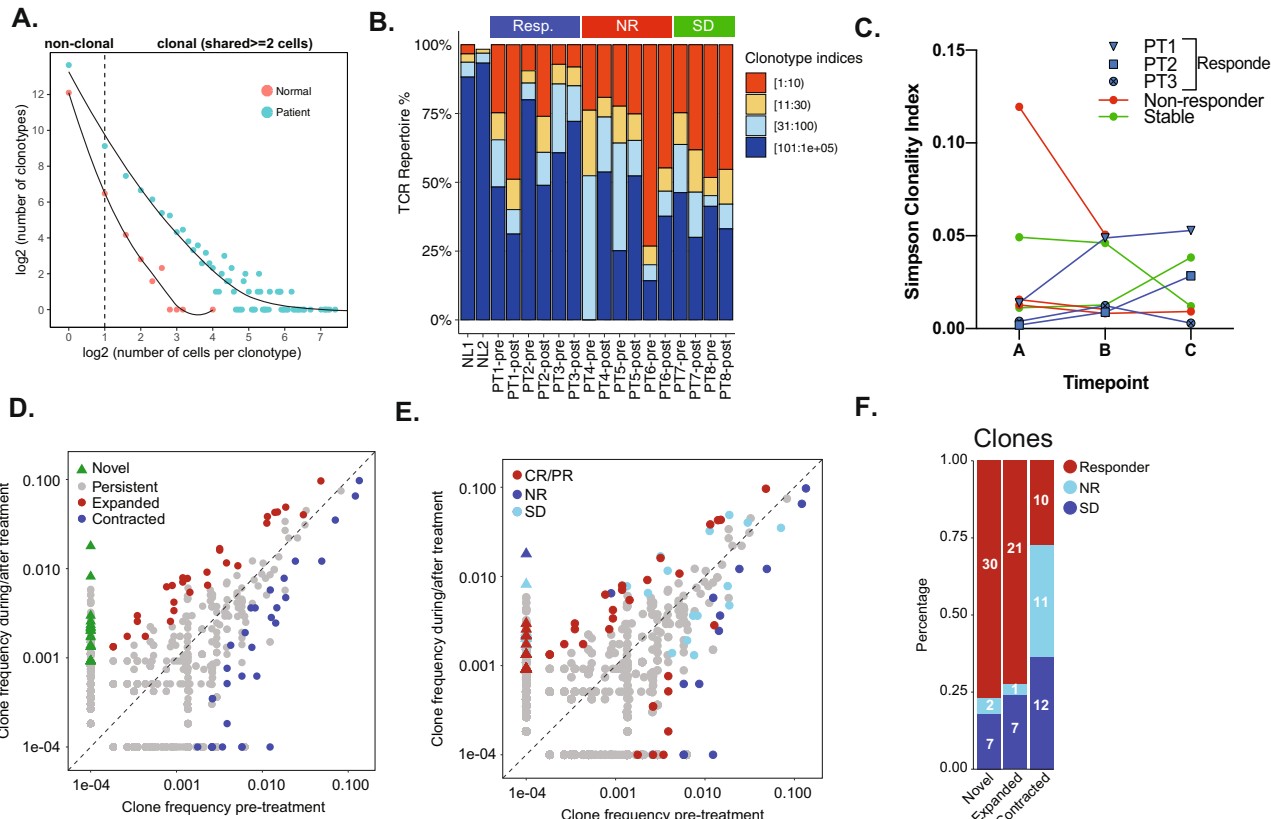

**Fig. 2 T-cell receptor clonotype assessment across different patients and timepoints. A** Scatterplot of the correlation between the number of T cell clonotypes and the size of the clonotype i.e. the number of T cells contributing to the clonotype. **B** Distribution of the most abundant clonotypes by patient. **C** Simpson's clonality index of individual patients at each of their respective timepoints. **D** Scatterplots of clonotypes change of post- versus pre-treatment and **E** by response groups. **F** Number of novel, expanded and contracted clonotypes by response group.

majority (70%) of significantly contracted clones were found in NR and SD patients (Fisher's exact $p < 0.05$) (Fig. 2F). These findings suggested that the capacity of TCR repertoire reactivity to treatment is variable, similar to what is seen in solid cancers[19].

**Heterogeneity in T cell populations of AML patients.** We next profiled the T-cell landscape to delineate distinct phenotypic groups contributing to the clonotypes. We identified 5 (2 conventional and 3 unconventional)[48] T-cell phenotypes in 25,798 T cells from 22 BM aspirates before and after treatment in AML patients (Fig. 3A). The 2 conventional phenotypes were CD4+ and CD8+ cells, constituting 53% and 35% of BM T cells at pretreatment, and 30.9% and 37.4% of BM T cells at posttreatment, respectively. The CD4:CD8 ratio of 1.51 in pretreatment BMs was lower than that in healthy BMs (1.88) and decreased further to 0.82 following treatment. The 3 unconventional T-cell phenotypes were gamma-delta (γδ) cells, mucosal associated invariant T-cells (MAIT) cells and all other (unconv T) cells, constituted 2.3%, 2.1% and 7% of T cells in pretreatment BMs, versus 8.5%, 14.4% and 8.5% of T cells in posttreatment BMs, respectively. Thus, proportion of CD8+, γδ and MAIT cells increase, while that of CD4+ cells decreased following treatment on aggregate across all patients (Fig. 3A). Further, there were marked variations in the distribution of the five T cell phenotypes among the 8 patients and at different timepoints of treatment indicating patient-specific T cell distributions, although some similar trends were also noted (Fig. 3B). For instance, at pretreatment, CD4+ cells were the most common cell type in responders (64.3%) and NR (42.48%) patients, whereas CD8+ cells were the most common in SD (53.48%) patients

(Supplementary Fig. 4A). Following treatment, CD8+ cells were the most common cell type in responders (30.28%) and SD patients (57.5%), whereas NR patients had persistently elevated CD4+ cells (52.91%) (Supplementary Fig. 4A). Further, γδ and MAIT cells increased in responders following treatment (Supplementary Fig. 4A), although this effect was primarily driven by PT1. These findings revealed heterogeneous T cell phenotypes across patients and between response groups, and reveal dynamic changes occurring longitudinally following treatment.

**Subclassification of T cells reveals distinct cellular phenotypes.** We then used multiparametric flow cytometry on pretreatment bone marrow samples in 33 patients (13 responders and 20 non-responders) conducted at time of enrollment in the azacitidine/nivolumab clinical trial and demonstrated significantly higher CD3+ ($p = 0.027$) and CD3+ CD8+ ($p = 0.044$) cells in responders (Supplementary Fig. 4A, B), consistent with a possible adaptive T-cell infiltration in responders. However, our clinical flow cytometry panel precluded the in-depth T-cell phenotyping that can be alternatively derived from scRNA analysis. We therefore further classified T cells from our scRNA profiling based on the expression of canonical genes (Fig. 3C and Supplementary Fig. 4C). CD4+ cells clustered into CD4+ naïve and CD4+ effector subsets including FOXP3+ ($T_{reg}$), T helper 1 cells ($T_H1$), $T_H17$ cells, and CD4+ cytotoxic (CTL) cells, while one cluster had no distinct expression profile (CD4NOS) (Fig. 3D). CD8+ clusters included CD8+ naïve, CD8+ STAT1 (enriched for STAT1 and expressed IFNγ pathway genes), CD8+ GZMK (enriched for GZMK expression), and CD8+ CTL (enriched for cytotoxic markers GZMB, GNLY and PRF1) (Fig. 3E). When examined

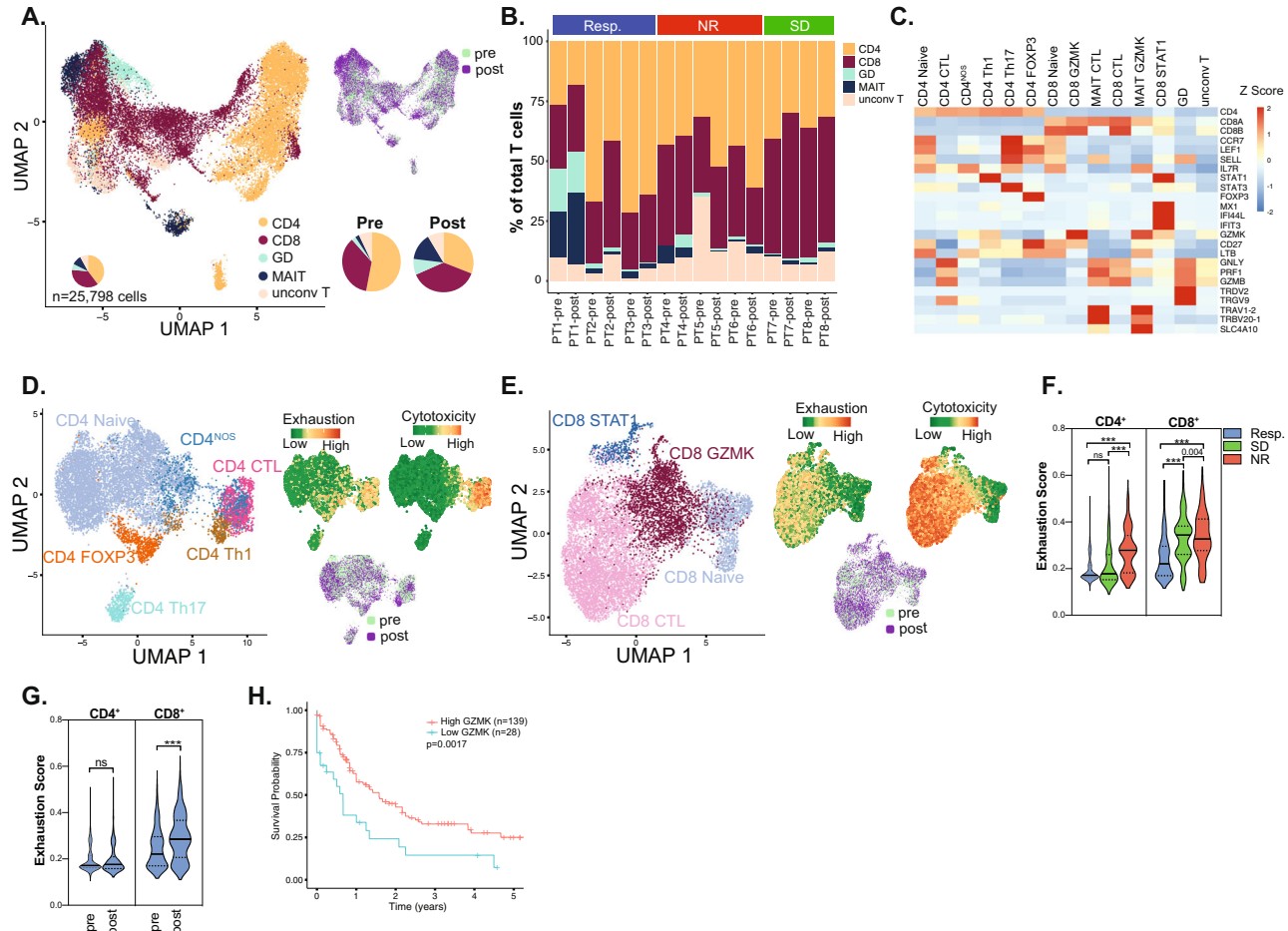

**Fig. 3 Characterization of T-cell subsets. A** UMAP of T cell subsets. **B** Distribution of T cell subsets prior to and following treatment. **C** Heatmap of canonical marker expression of identified T cell subsets. UMAP of the different **D** CD4+ and **E** CD8+ phenotypes with exhaustion and cytotoxicity scores projected onto the UMAP. **F** Mann–Whitney test for exhaustion scores of CD4+ and CD8+ T lymphocytes of different response groups prior to treatment. **G** Mann–Whitney test for exhaustion scores of CD4+ and CD8+ T lymphocytes at pre and post treatment in responders. **H** Overall survival of AML patients in TCGA cohort by GZMK expression.

across response groups, CD8+ GZMK constituted the majority of CD8+ cells in the 3 responders at pre-treatment and were significantly more abundant in responders compared to NR (mean of 53.2% in responders versus 18.8% in NR, $p = 0.03$) (Supplementary Fig. 4E, F). However, pretreatment CD8+ CTL cells were the least abundant cells in the 3 responders and were lower compared to the 3 non-responders (mean of 13.9% in responders vs 59.9% in NR, $p = 0.06$) (Supplementary Fig. 4C, D). Differential gene expression of CD8+ GZMK from responders compared to non-responders revealed significantly upregulated 137 genes which were enriched for tumor necrosis factor (TNF) α signaling pathway (Supplementary Fig. 4G and Supplementary Data 1), consistent an activated cell state in responders[49,50]. There were no discernible patterns for the changes in the CD4+ subsets at pre- or post-treatment across response groups in our single cell analysis (Supplementary Fig. 4H).

We next measured the cytotoxic and exhaustion scores of these cells by utilizing single cell gene set variation analysis (GSVA) for curated genes associated with these cell states[17,51,52]. Exhaustion scores in CD4+ and CD8+ cells increased with increasing cytotoxic scores, while pre- and post-treatment cells clustered together by cell phenotype (Fig. 3D, E). At pretreatment, exhaustion scores of CD4+ and CD8+ cells were lowest in the 3 responders compared to the 2 SD and 3 NR patients

($p < 0.0001$) (Fig. 3F). Following treatment, exhaustion scores of CD4+ were unchanged, while those of CD8+ significantly increased in the 3 responders ($p < 0.001$) (Fig. 3G). In non-responders, exhaustion scores decreased significantly ($p = 0.0005$) consistent with lower CD8+ CTL proportion following treatment (Supplementary Fig. 5A). These data suggested more dynamic changes in the CD8+ than CD4+ subsets following ICB-based treatment in AML especially in responders, with the most notable differences occurring in the pretreatment CD8+ GZMK and CD8+ CTL components.

Interpretation of these differences in CD8+ subsets among response groups should be tempered by the small sample size. However, the identification of GZMK as CD8+ cluster marker enriched in responders yet found in all patients warranted further investigation. GZMK expression was absent in AML cells (Supplementary Fig. 5B), while its expression in TME was distinctively in a subset of CD8+ cells, and less frequently in some NK, CD4+ and MAIT cells (Supplementary Fig. 5C, D). We thus investigated whether GZMK expression correlated with outcomes and found that AML patients in the TCGA cohort[53] with higher GZMK ($p = 0.0017$) expression had improved overall survival, suggesting that inherent immunity with elevated GZMK can elicit improved outcomes in AML (Fig. 3H). These findings warranted further characterization of CD8+ cells expressing GZMK.

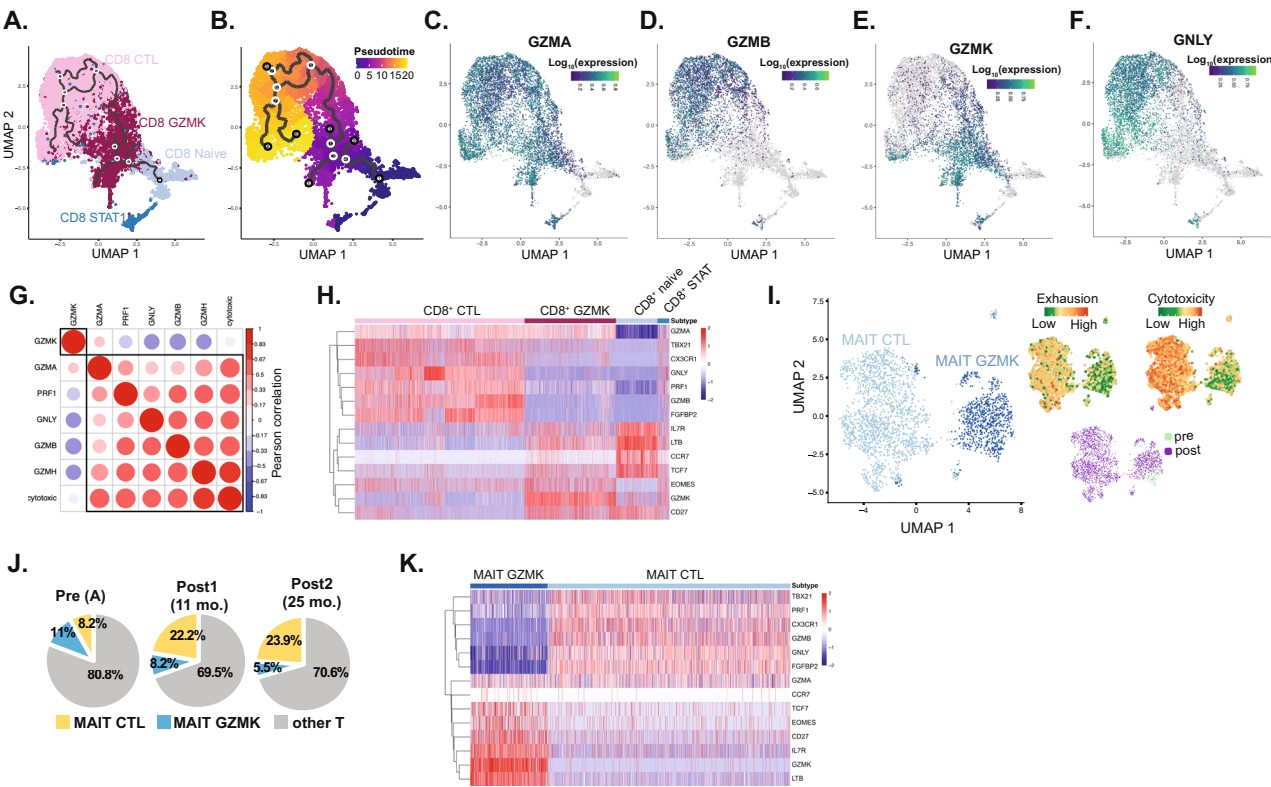

**Fig. 4 Trajectory analysis of CD8+ T cells. A**, **B** Monocle3-based pseudotemporal analysis of CD8+ subsets. **C–F** Expression of GZMA, GZMB, GZMK and GNLY in cells projected onto the trajectory of CD8+ continuum. **G** Pearson correlation between GZMK and other cytotoxic genes in CD8+ cells. **H** Heatmap of differentially expressed genes among CD8+ T lymphocyte subsets. **I** UMAP of MAIT cells with exhaustion and cytotoxicity scores projection. **J** Distribution of T-cell subsets in PT1. **K** Heatmap of differentially expressed genes between MAIT subsets.

**Pseudotemporal trajectory analysis revealed a CD8+ continuum**. There are 5 human granzyme genes (*GZMA*, *GZMB*, *GZMH*, *GZMM*, and *GZMK*) with the function of only GZMA and GZMB well described[54]. To investigate whether the expression of granzymes could reflect a distinctive marker for cell state program, we conducted pseudotemporal trajectory analysis and revealed a CD8+ continuum whereby CD8+ GZMK cells are intermediary to CD8+ naïve and CD8+ CTL cells (Fig. 4A, B). We also observed distinctive granzymes A, B, and K expression among the pseudotemporal axis of CD8+ cells. Specifically, *GZMA* was expressed ubiquitously in non-naïve CD8+ cells, whereas *GZMB* and *GZMK* expression profiles were expressed in different pseudotemporal trajectories (Fig. 4C–E). Specifically, *GZMB* was expressed at a later pseudotime in the cytotoxic T cells, while GZMK expression was prominent at an earlier pseudotime (Fig. 4C–E). The expression levels of the cytotoxic gene *GNLY* (delivers granzyme proteins into target cells for effector functions[54]) was diminished in GZMK-expressing CD8+ cells compared to CD8+ CTL cells (Fig. 4F). Further, there was negative correlation between *GZMK* and *GZMB*, *GNLY*, *PRF1*, *GZMH* and *PRF1* expression, as well as with cytotoxicity scores (Fig. 4G). Interestingly, the co-stimulatory genes *LTB* and *CD27*, the stem-like T cell transcription factor *TCF7*[10,28,51,55], and the T-cell memory transcription factor *EOMES*[11] were highly expressed in CD8+ GZMK cells, but not CD8+ CTL cells (Fig. 4H). To further explore the expression of GZMK in hematopoietic cells, we evaluated 3 different datasets from the Human Protein Atlas[56]. In 3 independent datasets from human blood cells[57–59], the highest expression of GZMK was noted to be in memory T-cells providing an independent validation of GZMK expression in memory T cells (Supplementary Fig. 5E–G). Based on the Human Protein Atlas[56], tonsillar tissue have the highest

expression of GZMK among lymphoid tissues. We therefore evaluated the expression of GZMK with the memory marker CD45RO and CD8 and found it to be co-expressed in subset of memory CD8 cells (Supplementary Fig. 5H–K). Further, recent studies demonstrated that GZMK distinguishes different subsets of memory T cells[60–62]. These findings, supported by the pseudotemporal trajectory analysis, suggested a continuum of CD8+ cells with an intermediary, distinctive, CD8+ GZMK population in AML that is characterized by high GZMK expression, and harbored stem-like and memory T cell markers.

**GZMK expression delineates MAIT subsets in PT1**. Unbiased clustering of MAIT cells revealed 2 distinct phenotypes: one enriched for less exhausted, GZMK-expressing cluster (MAIT GZMK) and another enriched for *GNLY/GZMB* cytotoxic genes (MAIT CTL), similar to CD8+ cells (Fig. 4I). Of note, 89.9% of MAIT cells in our analysis were contributed by PT1 (responder) who had a unique clinical course (Supplementary Fig. 6A). Briefly, PT1 had refractory AML to azacitidine (9/2015-4/2016), and to salvage with enasidenib (04/2016-07/2016) for *IDH2*-mutated refractory AML. At 88 years of age, he started a second salvage regimen with combined azacitidine/nivolumab with a partial response attained at 11 months from treatment initiation. He had sustained clinical benefit while on ICB-based therapy for 32 months until he developed ICB-induced pneumonitis necessitating switching treatment. Since this patient did not have a response to single agent azacitidine, then demonstrated a durable partial response to azacitidine/nivolumab, we postulate that the response was primarily driven by nivolumab. In PT1, the proportion of MAIT GZMK decreased from 11%, to 8.2% then 5.5%, while MAIT CTL increased from 8.2%, to 22.2% to 23.9% following treatment (Fig. 4J). Conclusions pertaining to MAIT cells

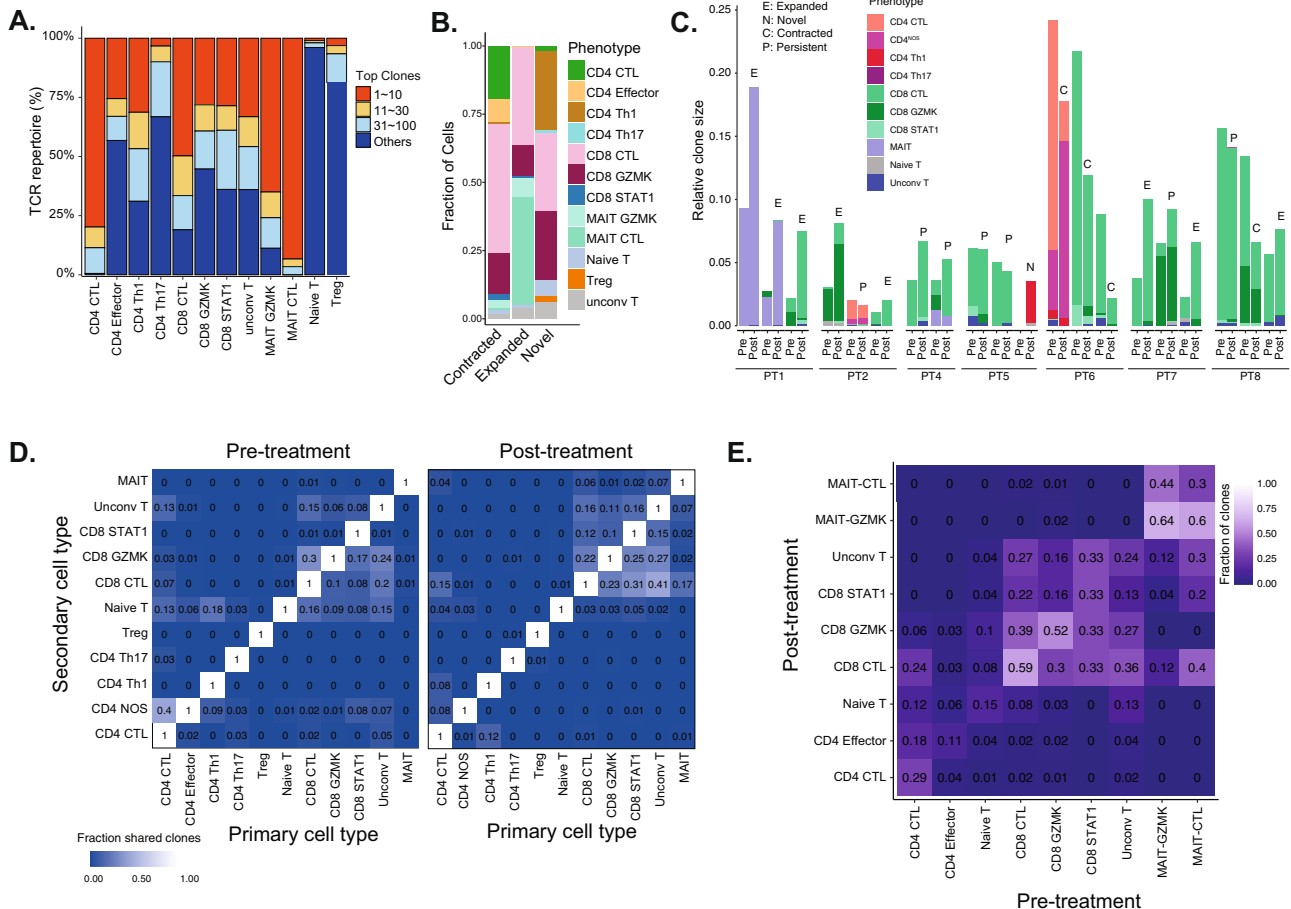

**Fig. 5 T-cell clonotype analysis. A** Distribution of the TCR clonotype frequency by cell type. **B** Number of novel, expanded and contracted clonotypes by cell type. **C** Fraction of T cells from top, most abundant 3 clonotypes. **D** Heatmap of overlapping clonotypes between different cell types at pre- and post-treatment timepoints. **E** Heatmap for observed phenotype transitions for matched clones at pre- and post- treatment timepoints.

are heralded by the derivation of MAIT cells from 1 patient only, however the interesting clinical course of this patient associated with MAIT cell expansion warranted highlighting these findings.

Similar to CD8$^+$ GZMK cells, MAIT GZMK cells were enriched for CD27, LTB, TCF7, and EOMES (Fig. 4K). Therefore, GZMK expression distinctly delineated subsets of CD8$^+$ and MAIT cells suggesting a unique transcriptional program correlated with GZMK expression. We therefore conducted gene expression profiling comparing GZMK and CTL subsets of each of CD8$^+$ and MAIT cells. Pathway enrichment of the 33 overlapping genes in the CD8$^+$ and MAIT GZMK versus CTL signatures demonstrated highest enrichment for pathways involved in leukocyte differentiation, calcium signaling, and cytokine production (Supplementary Fig. 6B). Importantly, calcium signaling regulates T cell differentiation and activation, is required for achieving T cell functional specificity, and regulates cytokine secretion and cytotoxic pathways[63].

**Phenotypic characterization of TCR clonotypes.** Building on the TCR profiling that revealed variable clonotype changes among patients (Fig. 2E–G), we leveraged the paired scTCR and scRNA profiling to integrate clonotype profiling with T-cell phenotypic states and then infer phenotypic activities[64]. Among T cells, cytotoxic subtypes had higher degree of clonal dominance (Fig. 5A). We next evaluated the contribution of the different T-cell subtypes to the clonotype pool following treatment. CD8$^+$ GZMK and CD8$^+$ CTL contributed the most to novel clones, whereas MAIT CTL had the highest fraction of expanded clones,

and CD8$^+$ CTL had the highest fraction of contracted clones (Fig. 5B). This suggests that CD8$^+$ cells in AML can indeed be reinvigorated to elicit clonotypic changes in response to therapy.

We next evaluated the T-cell phenotypes contributing to the top 3 most abundant clonotypes at pre- and post-treatment for each patient, similar to previous studies in skin cancers following PD-1 blockade[28]. The 3 most abundant clonotypes were largely contributed by cell types of same or similar phenotypes (Fig. 5C). For example, CD8$^+$ GZMK and CD8$^+$ CTL contributed to the most abundant clones of PT2, PT4, PT7 and PT8 (Fig. 5C). Also, MAIT GZMK and MAIT CTL of PT1 shared 2 of the most abundant clonotypes prior to treatment. Following treatment, the most abundant clonotypes expanded in 3/3 of the responders, but contracted in NR, while remaining unchanged in SD patients (Fig. 5C). Similar to what was seen in basal and squamous cell carcinoma following PD-1 treatment[28], there was no phenotypic instability of the dominants clonotypes in response to PD-1 blockade.

**Shared clonotypes among T cell lineages.** We next explored the lineage transition by investigating the fraction of clonotypes within a T-cell lineage (primary cell type) shared with other T-cell lineages (secondary cell type). We observed overlaps between CD8$^+$ subtypes, including CD8$^+$ GZMK and CD8$^+$ CTL, pre- and post-treatment (Fig. 5D). Moreover, the overlaps between CD8$^+$ subtypes were increased post-treatment, suggesting that the transitions between different activation states of CD8$^+$ cells were enhanced by PD-1 blockade (Fig. 5D). Specifically, the

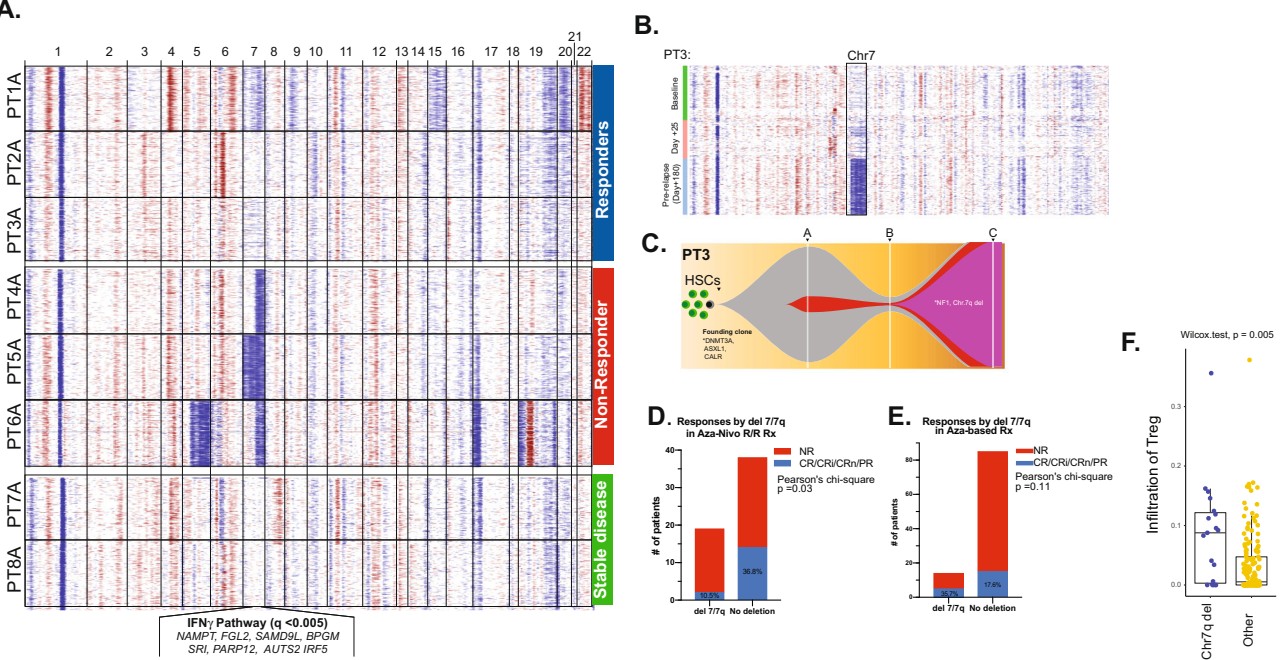

**Fig. 6 Correlation of responses with cytogenetics. A** Inferred copy number variation of representative 300 cells per patient at pretreatment (timepoint A). **B** Inferred copy number variation of the 3 timepoints (A, B and C) for PT3 (responder). **C** Fish plot of mutational evolution of PT3. **D** A two-sided Pearson's Chi-square for correlation analysis for responders to azacitidine/nivolumab and **E** azacitidine-based therapy based on chr7/7q deletion. **F** Mann–Whitney, two-sided test for CIBERSORTx analysis of $T_{reg}$ cell components in AML from TCGA by chr7/7q status ($n = 19$ for del7/7q and $n = 152$ for no deletion in chr7/7q). Center line represents the main and lower and upper hingest correspond to the first and third quartiles.

overlaps with the terminally effector CD8$^+$ CTL were highly increased following PD-1 blockade, including CD8$^+$ GZMK, suggesting a favored transition to CD8$^+$ CTL state. However, fewer overlaps were found for CD4$^+$ cell phenotypes pre- and post-treatment, suggesting limited transitions between CD4$^+$ cell states.

To assess clonotype stability, we evaluated the shared clonotype fraction for each T cell phenotype between pre- and post-treatment (Fig. 5E). MAIT, CD8$^+$ GZMK, and CD8$^+$ CTL cells demonstrated relative clonotype stability, whereas CD4$^+$ cells retained less clones post-treatment (Fig. 5E). Further, 30% of pretreatment CD8$^+$ GZMK clonotypes were shared with post-treatment CD8$^+$ CTL. These findings suggested shared clonotypes among closely related T-cell lineages that expanded following ICB therapy and thus an earlier antigen priming followed by phenotypic divergence.

**Chr7/7q loss is associated with resistance to ICB-based therapy.** While the T-cell landscape modulates responses to PD-1 blockade, tumor genomic alterations influence responses to immunotherapy in solid cancers[65–70]. However, AML cells harbor low tumor mutation burden[71], but large genomic alterations and SCNA that impact treatments and outcomes[53,72]. Therefore, we applied *inferCNV* tool[45] to deduce large-scale SCNAs of AML cells and was consistent with available clinical cytogenetics data for these patients from around the time of BM sampling (Supplementary Table 1). At pretreatment, 3/3 NR patients (PT4, PT5 and PT6) had inferred loss of chr7/7q in most of their malignant cells (Fig. 6A). Interestingly, PT3 (responder with CR) had an emergent inferred chr7q deletion (confirmed in 15/20 cells on karyotyping), which preceded the clinical and morphologic relapse (Fig. 6B, C) and was associated with loss of TCR clonal dominance (Fig. 2C).

To validate chr7/7q loss association with resistance to ICB-based therapy, we evaluated 57 R/R AML patients treated on

protocol with azacitidine and nivolumab who had evaluable pretreatment cytogenetic profiling. Interestingly, only 10.5% (2/19) of patients with chr7/7q deletion responded to treatment compared with 36.8% (14/38) of patients without deletion ($p = 0.03$) (Fig. 6D). To decouple azacitidine from nivolumab effect, we conducted a similar analysis on an independent cohort of R/R AML cohort ($n = 99$) treated on azacitidine-based clinical trials without ICB therapies. Interestingly, 35% (5/14) of R/R AML patients with chr7/7q loss achieved a complete or partial response compared with 17.6% (15/85) without the deletion ($p = 0.11$) (Fig. 6E). These data suggest that chr7/7q loss is associated with resistance to the combination of nivolumab/azacitidine, but not to azacitidine therapy without ICB.

We next aimed to delineate potential dysregulated molecular process and biological pathways associated with chr7/7q resistance to ICB-based therapy. mTORC1 signaling, glycolysis and interferon γ (IFNγ) Hallmarks pathways were highly enriched in differentially expressed genes on chr7q between patients with loss and intact chr7/7q (Supplementary Data 2 and Supplementary Fig. 7A). Further, IFNγ pathway genes, namely interferon-related developmental regulator (*IFRD1*) and the histone lysine methyltransferases *KMT2C* and *KMT2E* were among the significantly downregulated ones in our analysis. Gene set enrichment analysis on all chr7q genes that were also detected in our scRNA profiling also demonstrated significant enrichment for IFNγ pathway ($q < 0.0005$). Since IFNγ signaling shapes the immune BM microenvironment[73] and was enriched on genes from chr7/7q region, we estimated the absolute infiltration of T cells in AML cohort of TCGA in correlation with chr7/7q loss via CIBERSORTx[74]. $T_{reg}$ cells were significantly ($p = 0.005$) higher in AML patients with chr7/7q loss ($n = 19$) compared to those with intact chr7/7q ($n = 152$) (Fig. 6F). Of note, CD8$^+$ T cells absolute infiltration was higher in patients with chr7/7q deletion (Supplementary Fig. 7B), however CIBERSORTx deconvolution does not allow further phenotypic description of CD8$^+$

subsets. Additionally, TCGA AML patients with chr7/7q loss had significantly worse survival than patients without chr7/7q alteration ($p = 0.015$) (Supplementary Fig. 7C). Thus, chr7/7q loss appears to associate with or promote an immunosuppressive environment enriched in $T_{reg}$ immune cells and confers inferior outcomes.

## Discussion

Disentangling AML cells from their complex, immune-rich microenvironment can uncover the T-cell phenotypes in the AML milieu[1]. Assessing TCR repertoires can infer T-cell reactivity to treatment[19,64]. Paired analysis of both T-cell subsets and TCR repertoires provide a more accurate assessment of the functional properties of T cells in response to PD-1 blockade therapy[19–28]. In this study, we leveraged paired scRNA and scTCR profiling of AML BMs before and after treatment to understand the dynamics of T cells and their repertoires.

Our paired scRNA and scTCR profiling supported an adaptive T-cell repertoire that is primarily expanded by cytotoxic T-cell clonotypes emerging primarily in responders and SD patients following ICB-based therapy. In one patient (PT3), clonotype contraction and the loss of chr7/7q preceded relapse. Interestingly, some clonotypes were shared among different T-cell phenotypes. This suggests that while clonotypes may arise from same clone, T cells can become functionally divergent depending on microenvironment signals[64]. Also, it is possible that CD8$^+$ GZMK cells undergo functional differentiation into CD8 + CTL concurrent with loss of GZMK expression and acquisition of GZMB, which is supported by the trajectory analysis and the 30% overlap in TCR clonotypes at pre- and post-treatment (Fig. 5E). Of note, checkpoint blockade can indeed induce repertoire expansions in solid cancers[12,28,75]. However, in skin cancer, most of the expansions occurred in recruited novel clones. We found that in AML, both expanded and novel clonotypes, potentially representing locally and recruited clonotypes, respectively, are involved in the response. The difference compared to solid cancers could be related to the BM being a primary site for T-cell maturation thus affording PD-1 blockade rejuvenation of infiltrating T cells.

GZMB is well described in mediating apoptosis[76–79]. However, GZMK is considered an 'orphan' granzyme, with some studies considering it a cytolytic marker, while others suggesting no cytotoxic activity[35,54]. Our data support that GZMK identifies a CD8$^+$ subset enriched for stem-like and memory properties. GZMK cells had low expression of cytotoxic genes (*PRF1/GNLY/GZMB*) but harbored high expression of both transcription factors *TCF7* and *EOMES*. TCF7-expressing CD8$^+$ cells mediate and sustain immune responses via stem-like activities in a melanoma murine model treated with a PD-1 inhibitor, and following infection with lymphocytic choriomeningitis virus[80–82]. Additionally, CD8$^+$ TCF7$^+$ T cells define stem-like T cells in cancer patients, have enhanced self-renewal, multipotency, and give rise to more terminally differentiated effector CD8$^+$ T cells, further supporting our continuum model[55,83]. EOMES enables enrichment of memory CD8$^+$ T cells in the BM niche, promotes memory cell formation, and its expression in CD8$^+$ cells correlates with response to immunotherapy in melanoma[11,84,85]. Further, loss of *EOMES* result in defective memory T-cell populations, and its expression is associated with long-term memory-like T cells[86]. We further validated our findings by interrogating the human protein atlas datasets[57–59] and confirmed GZMK expression is highest in memory T cells. More recently, GZMK was shown to be differentially expressed in different subsets of memory T-cells and crucial for memory T cell function[60–62]. Collectively, these findings support that similar to

solid cancers, the T-cell memory subsets can influence responses to PD-1 blockade therapy in hematologic malignancies.

Our analyses allowed investigating genomic markers associated with resistance or response to checkpoint-blockade therapy. Tumor mutation burden and alterations in DNA damage response genes are associated with improved responses to checkpoint-blockade therapy in solid cancers[65–70]. Further, SCNAs are correlated with reduced response to immunotherapy in solid cancers[67,68,87]. However, AML has low mutation burden[71] but high cytogenetic alterations and SCNA[53,72], and thus investigating other genomic mechanisms that could influence the immune milieu is of value. AML patients with adverse cytogenetics have an immunosuppressive niche[2]. Here, we demonstrated that the loss in chr7/7q associated with resistance to ICB-therapy. While the underlying mechanisms are yet to be fully understood, these findings are important as they suggest that cytogenetic profiling, which is part of routine pathologic assessment for AML patients prior to treatment initiation, should be evaluated as a potential predictive biomarker for patient selection for ICB-based therapy trials in AML.

Currently, AML patients are only being treated with PD-1 therapy in the relapsed/refractory on clinal trials. There is a gap of knowledge in understanding T cell biology, subsets and TCR repertoires in response to treatment in this relevant treatment context. Our analyses, although conducted on a relatively small number of patients provides an in-depth look at T cell architecture and the adaptive T cell repertoire profiles in response to therapy. These data build a framework for deeper understanding of the molecular dynamics of T cells and AML cells, and can hopefully drive strategies to develop optimal personalized combinatorial approaches to improve efficacy and outcomes for these patients.

## Methods

**Human participants and treatment regimen.** Patients ≥18 years of age who had failed prior therapy for AML were eligible to participate on combined azacitidine and nivolumab trial (ClinalTrials.gov identifier: NCT02397720; full protocol is included in Supplementary Note 1). Bone marrows from 8 patients on NCT02397720 protocol and 2 healthy donors were characterized in this study. All patients had histologically proved relapsed or refractory acute myeloid leukemia. Treatment consisted of azacitidine 75 mg/m$^2$ days 1 to 7 administered intravenously (i.v.) over 60 to 90 min or subcutaneously, and nivolumab 3 mg/kg administered as a 60 to 90 min i.v. infusion on days 1 and 14 of each cycle. Each cycle consisted of 28 days and were repeated every 4 to 6 weeks, depending on count recovery. Patients continued on therapy as long as they had evidence of clinical benefit. Response assessment was conducted using European Leukemia Network (ELN) response criteria[72]. A written informed consent for enrollment on the protocol and for all uses of human material was approved by the internal review board of University of Texas M D Anderson Cancer Center was obtained. The study was conducted in accordance with the Declaration of Helsinki.

**Sample collection and processing.** Bone marrow (BM) biopsies were routinely collected prior to treatment initiation and at different timepoints for response assessments. BM samples were freshly frozen with freezing medium containing 20% fetal calf serum (FCS) and 10% DMSO in Dulbecco's Modified Eagle Medium (DMEM), and stored in liquid nitrogen.

**Cells preparation for single cell profiling.** All BM samples stored in liquid nitrogen were retrieved right before sample processing. To maximize the cellular viability recovery, samples were processed in batches according to in house developed protocol and 10x Genomics Demonstrated Protocol Cell Preparation Guide (Document CG00053). Briefly, cells were gently thawed in water bath at 37 °C until partially thawed and immediately placed on ice. Next, cells were gently transferred to a 10 ml media (10 ml alphaMEM + 20%FCS) and centrifuged (453 g for 5 min). After removal of supernatant, the cell pellet was carefully resuspended in 10 ml enriched media (alphaMEM + 20% FCS supplemented with 500 μl Heparine (Cat# 9041-08-1; Alfa Aesar), 15 μl DNase (Cat# 89835, thermo Fischer Scientifc) and 500 μl MgSO$_4$), followed by incubation in 37 °C for 15 min. After incubation, cells were centrifuged and gently washed twice in 1.5–3 ml of 0.04% BSA in PBS. Additionally, cells were passed through 35μm strainer (BD Falcon® 5 ml Round-Bottom Tubes with Cell Strainer Cap, Corning, NY, USA) to eliminate cell clumps. Next, cells were stained with 0.4% Trypan blue and quantified and

assessed for viability using the cell automated counting machine Cellometer Mini (Nexcelom, Lawrence, MA, US), as well as using standard hemocytometer and light microscopy.

**Library preparation for 10x Genomics 5′ single cell RNA and V(D)J sequencing**. The 5′ gene expression libraries (5′GEX) and scTCR enriched libraries were prepared using the 10x Single Cell Immune Profiling Solution (https://www.10xgenomics.com/products/single-cell-immune-profiling/), according to the manufacturer's protocol CG000086 Chromium Single Cell V(D)J Reagent Kits User Guide Rev G (v1 Chemistry), (10x Genomics, Pleasanton, CA, USA). This method did not require any custom primers, and all primers necessary for cDNA generation and preparation of 5′ GEX libraries and scTCR libraries were supplied within 10x Genomics reagents. The brief description of the key steps conducted for libraries generation are summarized below.

**cDNA generation from single cell**. the single cell suspensions with a targeted cell recovery of 10,000 cells per sample were mixed with Master Mix and loaded into the Chromium Chip A along with the barcoded Single Cell VDJ 5′ Gel Beads v1 and Partitioning Oil. The nanoliter-scale Gel Beads-in-emulsion (GEMs) were generated using 10x Chromium Controller. Next, GEMs were captured and incubated (Step 1: 53 ℃ for 45 min, Step 2: 85 ℃ for 5 min) to produce 10x Barcoded with unique molecular index (UMI), full-length cDNA from polyadenylated mRNA in reverse transcription reaction. Following steps included breaking the GEMs, Post GEM-RT Cleanup and PCR amplification of released cDNA with primers against common 5′ and 3′ ends added during GEM- RT (Step 1: 98 ℃ for 45 s, Step 2: 98 ℃ for 20 s, Step 3: 67 ℃ for 30 s, Step 4: 72 ℃ for 1 min, Step 5: 72 ℃ for 1 min, Hold: 4 ℃; Steps 2–4 were performed in total of 13 cycles). The obtained cDNA was cleaned-up using SPRIselect Reagent Kit (Beckman Coulter, Brea, CA, USA) and quantified using an Agilent 4200 Tape Station HS D5000 Assay (Agilent Technologies, Santa Clara, CA, USA). Such prepared cDNA was further used to generate 5′GEX libraries and TCR-enriched libraries.

**Construction of 5′GEX libraries**. 50 ng mass of cDNA of each sample was used. The main steps of 5′GEX library preparation included: (1) fragmentation, end repair and A-tailing followed by double sided size selection (2) adapter ligation with post-ligation cleanup, and 3) sample index PCR followed by the double-sided cleanup and QC. The enzymatic fragmentation and size selection (SPRIselect Reagent Kit, Beckman Coulter, Brea, CA, USA) were used to optimize the cDNA amplicon size followed by adding P5 and P7 sample indexes (used in Illumina sequencers), and Illumina R2 sequence via processes of end repair, A-tailing (Hold: 4 ℃, Step 1: 32 ℃ for 5 min, Step 2: 65 ℃ for 30 min, Hold: 4 ℃), adapter ligation (20 ℃ for 15 min, Hold: 4 ℃) and Sample Index PCR (Step 1: 98 ℃ for 45 sec, Step 2: 98 ℃ for 20 sec, Step 3: 54 ℃ for 30 s, Step 4: 72 ℃ for 20 s, Step 5: 72 ℃ for 1 min, Hold: 4 ℃; Steps 2–4 were performed in total of 14 cycles) using Chromium i7 Sample Index (PN-220103, 10x Genomics, Pleasanton, CA, USA).

**TCR enrichment step**. 5 μl of cDNA of each sample was used. The TCR transcripts were enriched by two rounds of PCR amplification (Step 1: 98 ℃ for 45 sec, Step 2: 98 ℃ for 20 s, Step 3: 67 ℃ for 30 s, Step 4: 72 ℃ for 1 min, Step 5: 72 ℃ for 1 min, Hold: 4 ℃; Steps 2–4 were performed in total of 10 cycles) with primers specific to the TCR, with sample cleanup between each run of PCR. Noteworthy, the P5 was also added during enrichment. The double-sided size selection was performed on enriched target followed by QC step.

**Construction of TCR enriched libraries**. 50 ng mass of TCR enriched samples were used. The main steps of the process included: (1) fragmentation, end repair and A-tailing (Hold: 4 ℃, Step 1: 32 ℃ for 2 min, Step 2: 65 ℃ for 30 min, Hold: 4 ℃), (2) adapter ligation (20 ℃ for 15 min, Hold: 4 ℃) with post-ligation cleanup, (3) sample index PCR amplification (Step 1: 98 ℃ for 45 sec, Step 2: 98 ℃ for 20 s, Step 3: 54 ℃ for 30 s, Step 4: 72 ℃ for 20 s, Step 5: 72 ℃ for 1 min, Hold: 4 ℃; Steps 2–4 were performed in total of 9 cycles) using Chromium i7 Sample Index (PN-220103, 10x Genomics, Pleasanton, CA, USA), followed by the double-sided cleanup and QC step.

**Sequencing**. All libraries were checked for the fragment size distribution using Agilent 4200 Tape Station HS D5000 Assay (Agilent Technologies, Santa Clara, CA, USA) and quantified with Qubit Fluorometric dsDNA Quantification kit (Thermo Fisher Scientific, Waltham, MA, US). Each of 5′GEX and TCR enriched libraries were prepared with unique indexes allowing for multiplexing. The 5′GEX libraries were sequenced each of on a separate lane of HiSeq4000 flow cell (Illumina, San Diego, CA, USA) to target sequencing depth of 50,000 read pairs per sample, in total of four batches. All TCR enriched libraries were equimolarly pooled and sequenced in one sequencing run using NovaSeq6000 S2-Xp 100 (Illumina, San Diego, CA, USA) as required lower sequencing depth of 5,000 read pairs per sample. All libraries were sequenced at the MDACC Advanced Technology Genomics Core (ATGC, Houston, TX, USA) facility under 10x Genomics recommended cycling parameters for 5′GEX libraries (Read 1–26 cycles, Read 2–98

cycles; with few exceptions for samples sequenced on the same flow cell along with scATAC libraries: Read 1–100 cycles, Read 2–100 cycles) and for scTCR enriched libraries (Read 1–26 cycles, Read 2–91 cycles), with sequencing format of 100nt.

Raw sequencing data processing, quality check, data filtering, doublets removal, batch effect evaluation and data normalization: The raw scRNA-seq data were pre-processed (demultiplex cellular barcodes, read alignment, and generation of gene read count matrix) using Cell Ranger Single Cell Software Suite provided by 10x Genomics. Detailed QC metrics were generated and evaluated. Genes detected in fewer than 3 cells and cells with low complexity libraries (less than 200 genes were detected) were filtered out and excluded from subsequent analysis. Low-quality cells where >15% of transcripts derived from the mitochondria were considered apoptotic and also excluded. Following the initial clustering, we removed likely cell doublets from all clusters. Doublets were identified by the following methods[88] (1) library complexity-cells are outliers in terms of library complexity. (2) Cluster distribution- doublets or multiplets likely form distinct clusters with hybrid expression features and exhibit an aberrantly high gene count; (3) cluster marker gene expression–cells of a cluster express markers from distinct lineages (e.g., cells in the T-cell cluster showed expression of epithelial cell markers; 4) Non-T cells expressing TCR or Non-B cells expressing BCR. After these steps, we evaluated the doublet score using DoubletFinder[44] and confirmed that most doublets should have been appropriately removed (Supplementary Fig. 2F, G).

Unsupervised cell clustering, dimensionality reduction using t-SNE and UMAP, and cluster relationship analysis: Library size normalization was performed in Seurat[89] on the filtered gene-cell matrix to obtain the normalized UMI count as previously described[36]. Then, the normalized gene-cell matrix was used to identify highly variable genes for unsupervised cell clustering in Seurat. The elbow plot was generated with the ElbowPlot function of Seurat and based on which, the number of significant principal components (PCs) were determined. Different resolution parameters for unsupervised clustering were then examined in order to determine the optimal number of clusters. For visualization, the dimensionality was further reduced using Uniform Manifold Approximation and Projection (UMAP)[90] methods with *Seurat* function *RunUMAP*. In addition, *Monocle 3* alpha (http://cole-trapnell-lab.github.io/monocle-release/monocle3/)[91] was applied as an independent tool for unsupervised clustering analysis (function *cluster_cells*) and UMAP was used by default with the *Monocle* functions *reduce_dimension* and *plot_cells* for dimensionality reduction and visualization of the *Monocle* clustering results. *Monocle 3* alpha was also used to construct the single-cell trajectories. The function *learn_graph* was run with default parameters. Batch effects was corrected using the R package HARMONY before clustering analysis in Seurat and using the *Alignment* function in Monocle3 before trajectory analysis.

**Determination of major cell types and cell states**. To define the major cell type of each single cell, differentially expressed genes (DEGs) were identified for each cell cluster using the *FindAllMarkers* analysis in the Seurat[89] package and the top 50 most significant DEGs were carefully reviewed. In parallel, feature plots were generated for top DEGs and a suggested set of canonical immune and stromal cell markers, a similar approach as previously described[10,14], followed by a manual review process. Enrichment of these markers (e.g., *PTPRC* for immune cells; *CD3D/E* for T cells; *CD8A/B* for CD8 T cells, *IL7R/CD4/CD40LG* for CD4 T cells; *CD19/MS4A1/CD79A* for B cells, etc.) in certain clusters was considered a strong indication of the clusters representing the corresponding cell types. The two approaches are combined to infer major cell types for each cell cluster according to the enrichment of marker genes and top-ranked DEGs in each cell cluster, as previously described[10]. In further confirm our cell type annotation, data from the two normal bone marrow samples were also mapped to multimodal human BMNC reference dataset provided by Seurat v4. Cell types of both reference mapping and inhouse annotation were reduced to 11 general cell type groups (B, Monocytes, CD4, CD8, DC, NK, Unconventional T, Progenitors, HSC, Erythroid and Plasma). Concordance rate was calculated by dividing the number of cells with consistent cell type group annotation between reference mapping and inhouse annotation by the total number of cells tested. Concordance rates were also calculated for patients' data after cell type assignment.

**Infer large-scale copy number variations (CNVs)**. The tool *inferCNV* (https://github.com/broadinstitute/inferCNV) was applied to infer the large-scale CNVs from scRNA-seq data and monocytes from normal bone marrow of this dataset were used as a control for CNV analysis. Initial CNVs were estimated by sorting the analyzed genes by their chromosomal locations and applying a moving average to the relative expression values, with a sliding window of 100 genes within each chromosome, as previously described[46]. Finally, malignant cells were distinguished from normal cells based on information integrated from multiple sources including marker genes expression, inferred large-scale CNVs, and their cluster distribution per patient with cells from normal bone marrow.

**Pathway enrichment analysis**. For pathway analysis, the curated gene sets (including Hallmark and KEGG gene sets) were downloaded from the Molecular Signature Database (MSigDB, http://software.broadinstitute.org/gsea/msigdb/index.jsp), single-sample GSVA (ssGSVA) was applied to the scRNA-seq data and

pathway scores were calculated for each cell using *gsva* function in GSVA software package[52]. Heatmaps were generated using the *heatmap* function in pheatmap R package for filtered DEGs.

TCR V(D)J sequence assembly, clonotype calling, TCR diversity and clonality analysis and integration with scRNA-seq data: Cell Ranger v3.0.2 for V(D)J sequence assembly was applied for TCR/BCR reconstruction and paired clonotype calling. The CDR3 motif was located and the productivity was determined for each single cell. The clonotype landscape was then assessed and the clonal fraction of each identified clonotype was calculated. The TCR clonotype diversity matrix was calculated using the tcR R package[92]. The clonotype data was then integrated with the T-cell phenotype data inferred from single cell gene expression analysis based on the shared cell barcodes.

**Immune cell deconvolution of TCGA sample**. CIBERSORTx[74] was applied to the normalized bulk RNA-seq data with the LM22 gene signature to estimate the absolute level of immune infiltrations.

**Statistical analysis**. Statistical differences were calculated with an unpaired Student's *t*-test for 2-tails. Spearman's correlation coefficient was used. A value of $p < 0.05$ was considered statistically significant. For multiple *t*-tests, false discovery rate with $q < 0.05$ was used for statistical significance. All statistical analysis were performed using packages in R (R Foundation for Statistical Computing) and in GraphPad Prism Software version 8.

**Multiplex immunofluorescence**. Chromogen-based immunohistochemistry and multiplex immunofluorescence using CD8, Granzyme K (GZMK), and CD45RO were previously optimized and validated. Each antibody was assessed by a uniplex IF assay to generate spectral libraries required for multiplex IF image analysis. Uniplex IF staining was performed using the Opal 7 kit (catalog #NEL797001KT; Akoya Biosciences, Marlborough, MA), which an individual tyramide signal amplification (TSA)-conjugated fluorophores is used to detect various targets within an IF assay. The process consists in an automatized process in an autostainer (Leica Bond RX, Leica Biosystems, Vista, CA), which performs the following process: deparaffinization, thermoregulation (boiling and cooling), protein stabilization, incubation, antigen retrieval, and counterstaining with DAPI. For each antibody, a previous determined specification was used according the validation: CD8 (clone c8/144B, Thermo Scientific, 1:25), GZMK (polyclonal, Sigma Aldrich, 1:100), and CD45RO (clone UCHL1, Leica Bond, ready to use). The slides were imaged using the Vectra Polaris spectral imaging system (Akoya Biosciences, Marlborough, MA) using the fluorescence protocol at 10 nm λ from 420 nm to 720 nm, to extract fluorescent intensity information from the images. A similar approach was used to build the spectral library using the InForm 2.4.8 image analysis software (Akoya Biosciences, Marlborough, MA). In the multiplex IF slides, each batch was scanned with the Vectra Polaris imaging system using a reactive human tonsils as controls to calibrate the spectral image protocol. After low magnification scanning at ×10 using Phenochart 1.0.9, we selected the whole regions of interest (interfollicular areas). The high magnification areas were acquired using high resolution (×20) and accessed by InForm 2.4.8 software.

**Flow cytometry**. Fresh bone marrow aspirate is collected in EDTA tubes. Flow cytometry is performed within 24 h using CD45-V500 (BD Biosciences Cat # 560777, Clone HI30) and CD34-PE-Cy7 (BD Biosciences Cat # 348791, Clone 8G12). 5 ul (1:1 dilution) of each antibody was added to each tube which contains about 200,000 cells then flow cytometry analysis was run on BD FACSCanto 8-color and 10-color instruments and analyzed on FCEXPRESSION V6.

**Reporting summary**. Further information on research design is available in the Nature Research Reporting Summary linked to this article.

## Data availability
The sequencing data can be accessed from European Genome-phenome Archive (EGA) database (https://ega-archive.org/studies/EGAS00001004894) with EGA ID number for 5' scRNA-seq (EGAD00001007672), scTCR-seq (EGAD00001007674) and DNA-seq (EGAD00001007671) data. All other data including the source data for main and supplementary figures are also available in the Source File. Source data are provided with this paper.

## Code availability
All codes used in this manuscript are based on public library packages that are listed in the Methods section.

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

## Acknowledgements

We would like to acknowledge the Advanced Technology Genomics Core (ATGC) at M D Anderson and 10xgenomics for sequencing and technical support. The work was funded in part by T32 NIH fellowship to H.A.A.; CPRIT MIRA (RP160693) and Department of Leukemia SPORE (5 P50 CA100632-09) to S.M.K.; CA016672 and NIH 1 S10OD024977-01 to Advanced Technology Genomics Core (ATGC) and MDACC start-up to K.R.; MDACC start-up to L.W.; CPRIT MIRA (RP160693) and MD Anderson Cancer Center Leukemia SPORE CA100632 to S.K.; the MD Anderson Cancer Center Support Grant (CCSG) CA106672, the MD Anderson Cancer Leukemia SPORE CA100632, the Dick Clark Immunotherapy Fund and generous philanthropic contributions to the MD Anderson Moon Shots Program to N.D.; APOLLO, Welch Foundation and CPRIT grants to A.F.

## Author contributions

Conceived the project: H.A.A., N.D. and A.F. Lead library preparation and experimental optimization: K.To., P.B. and K.R. Lead computational analysis: D.H. and L.W. Optimized experiments, analyzed the data, wrote the manuscript: H.A.A. D.H., K.To., L.W., K.R., N.D. and A.F. Contributed to data analysis, study design, experimental design, optimization and/or writing the manuscript: J.S.I., P.K.R., Z.A., W.W., H. C.B. R.W., F.W., G.A., M.R.G., K.Ta., L.L., J.Z., S.B. Immunostaining: M.L.M.P., L.M.S., E.R.P., W.L., and A.,T. Bone marrow processing and clinical annotations: G.A., J.T.M., Z.A., S.M.K. Statistical tests and design: J.N. and M.D. Contributed conceptually to data analysis, writing and design: M.K., G.G.M., P.S. and J.P.A., S.M.K. Hematopathology: W.W.

## Competing interests

K.Ta. reports consulting and advisory roles for Symbio Pharmaceuticals, Novartis, GSK and Celgene/BMS. G.A. reports consulting fees from Novartis and Poseida therapeutics, and research funding from Merck and Jenssen Pharmaceuticals. M.R.G. reports consulting fees with VeraStem Oncology and stock/ownership interest KDAc Therapeutics. M.K. reports grant support and consulting fees from AbbVie, Genentech, F. Hoffmann La-Roche, Stemline Therapeutics, Forty-Seven, consulting fees from Amgen and Kisoji, grant support from Eli Lilly, Cellectis, Calithera, Ablynx, Agios, Ascentage, AstraZeneca, Rafael Pharmaceutical, Sanofi, royalties and stock options from Reata Pharmaceutical Inc. P.S. reports consulting, advisory roles, and/or stocks/ownership for Achelois, Adaptive Biotechnologies, Affini-T, Apricity, BioAtla, BioNTech, Candel Therapeutics, Catalio, Codiak, Constellation, Dragonfly, Earli, Enable Medicine, Glympse, Hummingbird, ImaginAb, Infinity Pharma, Jounce, JSL Health, Lava Therapeutics, Lytix, Marker, Oncolytics, PBM Capital, Phenomic AI, Polaris Pharma, Sporos, Time Bioventures, Trained Therapeutix, Two Bear Capital, Venn Biosciences. J.P.A. reports consulting, advisory roles, and/or stocks/ownership for Achelois, Adaptive Biotechnologies, Apricity, BioAtla, BioNTech, Candel Therapeutics, Codiak, Dragonfly, Earli, Enable Medicine, Hummingbird, ImaginAb, Jounce, Lava Therapeutics, Lytix, Marker, PBM Capital, Phenomic AI, Polaris Pharma, Time Bioventures, Trained Therapeutix, Two Bear Capital, Venn Biosciences. N.D. reports research funding from Daiichi Sankyo, Bristol-Myers Squibb, Pfizer, Karyopharm, Sevier, Genentech, Astellas, Abbvie, Genentech, Novimmune, Amgen, Trovagene, Gilead and ImmunoGen and has served in a consulting or advisory role for Daiichi Sankyo, Bristol-Myers Squibb, Pfizer, Novartis, Celgene, AbbVie, Genentech, Servier, Trillium, Syndax, Trovagene, Astellas, Gilead and Agios. The remaining authors declare no competing interests.
