## [Peer Review File · Nature Communications]

Single cell T cell landscape and T cell receptor repertoire profiling of AML in context of PD-1 blockade therapyREVIEWER COMMENTS

Reviewer #1 (Remarks to the Author):

Abbas et al. use a state-of-the-art combination approach of single cell gene expression profiling and single cell TCR sequencing to assess the T cell phenotype and clonotype in BM cells from AML patients treated with anti-PD-1 (nivolumab) plus azacitidine combination therapy at a single cell level. BM samples were derived from 8 patients enrolled in the NCT02397720 study of nivolumab+azacitidine in r/r AML and 2 healthy donors. Paired scRNA and scTCR profiling revealed changes in cellular subset composition and TCR clonotypes pre- and post nivolumab+azacitidine treatment. They found that the most abundant clonotypes expanded primarily in patients that responded or had stable disease while the most abundant clonotypes retracted in non-responders. Furthermore, the authors suggest GZMK expression in CD8 T cells and MAIT cells as a potential marker for improved outcome in AML. Finally, they identified that chromosome 7/7q loss was associated with resistance to anti-PD-1/azacitidine treatment in a larger cohort of 57 patients. While this study provides an in-depth look at T cell phenotypes and TCR repertoires in patients before and after anti-PD-1 based therapy, one major concern is that it remains at a merely descriptive level and I am not not convinced that it will be practice changing in the clinic. In addition, a major part of the results is based on quantitative assessment of cell population proportions, which has to be viewed with caution because of the limited number of patients assessed. While differences in gene signatures can be quantitatively assessed using scRNA-seq, mere quantification of cell populations should preferably be analyzed with other methods that allow for larger patient and cell numbers, e.g. flow cytometry.

Major Comments

- The authors should be cautious with claiming quantitative changes in certain cell types (e.g. CD4 vs. CD8 cell subset) due to the very low patient numbers per group. It cannot be ruled out that single patients with high cell numbers drive specific changes. Single cell sequencing should be rather utilized to compare changes in gene expression in specific cell subsets between the patient groups, e.g. did T cell effector signatures change in the CD8 subset in response to treatment. Quantitative changes in cell subset proportions should rather be confirmed by flow with a) larger patient numbers and b) larger cell numbers, to be able to statistically assess these changes.
- Why is timepoint C not presented in Figure 2C? Since this is a merely descriptive depiction of the clonality, all timepoints should be considered and reported. Does the time point "post" in Figure 2B and 3B refers to time point B or C – in particular in responders?
- Did decrease in clonality at timepoint C predict subsequent relapse? Did patients with persistent increase in T cell clonality stay in remission longer?
- In Figure 3G the text implies that only responders are depicted. How did the exhaustion scores change in non-responders?
- Pseudotime analysis revealed that GZMB expression was higher at a later pseudotime. Did GZMB expression correlate with exhaustion signature?
- The findings of co-expression of TCF7 and EOMES in the GZMK+ CD8 T cells, as well as their stem cell like properties are interesting, but rather descriptive. The manuscript would be improved by confirming these findings by flow and in functional assays.
- The MAIT cell data are interesting but have to be viewed with caution, since 90% of the cells were contributed by a single patient.
- The authors identify chr7/7q deletion as a resistance mechanism to checkpoint blockade and attribute this to expansion of regulatory T cells and reduced IFN γ signaling. What were the percentages/phenotype of Tregs in the patients in this study at different time points? Were these correlated with the duration of response/survival? Also here, functional assays to explore the role of IFN γ signaling in response to azacitidine/nivolumab would be highly beneficial.

Minor Comments

- Figure 2B and 3B would be clearer if the response of the individual patients was indicated as in Figure 6A.
- Please carefully check for typos:
 - o p.7 "... in clinical trial settings"
 - o p.2 "... stem-like properties likely." This sentence ends openly.

o p.10 "...expression levels of the cytotoxic gene GNLY, which delivers..."

Reviewer #2 (Remarks to the Author):

In this manuscript, the authors performed single cell RNA-sequencing on a large number of bone marrow cells from patients with relapsed/refractory AML both pre- and post-treatment alongside healthy donors. They also profiled T-cell receptors in those same cells. In summary, the authors discovered drug response and cell type proportion heterogeneity within the T-cells, along with variable TCR repertoires between responders and non-responders. Their analysis found a developmental continuum of T-cells with differing levels of the GZMK gene as well as a potential mechanism for resistance to PD-1 blockade therapy. Overall, the authors effectively use biological and computational techniques in this manuscript to evaluate their hypothesis. The manuscript is clearly written, and the results novel and of value to the scientific community.

1. The author's state: "We identified 5 (2 conventional and 3 unconventional)48 T cell phenotypes in 25,798 T cells from 22 BM aspirates before and after treatment in AML patients (Fig. 3A). The 2 conventional phenotypes were CD4+ and CD8+ cells, constituting 53% and 35% of BM T cells at pretreatment, and 30.9% and 37.4% of BM T cells at posttreatment, respectively." This section goes further to highlight cell type proportion differences between highly similar cell types, such as CD8 and MAIT cells. I do not believe that bioinformatics tools in their current state are accurate enough in cell type identification to allow for definitively stating cell type proportion differences that are only moderately altered between the two patient groups. This section could be further supported using experimental methods to verify these proportion differences in some samples.
2. In a couple of places in the manuscript the authors acknowledge the heterogeneity in the patient population. One case is the following: "Of note, 89.9% of MAIT cells in our analysis were contributed by PT1 (responder) who had a unique clinical course." Given such a dramatic patient-specific effect, how are the authors assessing the significance of their results?
3. In the Chr7/7q section, the authors found 13 significantly downregulated genes on chr7q in AML cells and used these genes to perform gene set enrichment analysis. This section could be improved by relaxing the significance criteria and performing the analysis with a larger set of genes in addition to the original analysis, as 13 is a very small gene set.
4. The author's state that "All datasets generated during and/or analyzed during the current study will be available from the corresponding authors." Will these data become publicly available upon publication?
5. Doublets were identified using robust criteria, which should remove doublets effectively, but new computational tools for doublet detection, such as Scrublet, DoubletDecon, and Solo, have shown very high accuracy in biologically validated datasets. Given the high transcriptional similarity between the T-cell populations of interest and the difficulty in identifying doublets because of this with the given techniques, I would suggest using a computational approach for further validation.
6. The legend for Figure 1C refers to a UMAP-based pseudotemporal trajectory analysis, but figure shows only the UMAP and cell type proportion predictions with no trajectory analysis.
7. In Figure 1D, my immediate concern with the green and red UMAP was potential batch effects, given the separation between the PT1C and Normal BM cells in non-malignant cell types. Upon further reflection, this could simply be an overplotting issue. Using side-by-side UMAPs in the same dimensionality reduction space would better highlight the integration and differences and alleviate batch effect concerns.
8. For Figure 3A, within each cell type do the distinct subclusters align with the mutation breakdown? If not, what transcriptional change causes the small subsets of cells to distinctly separate from the larger cell population?

Reviewer #3 (Remarks to the Author):

The authors analyze single-cell transcriptomics data from serial bone marrow samples of 8 patients treated for relapsed/refractory AML with azacytidine and the checkpoint inhibitor nivolumab. With

respect to immune cells, transcriptomic data are analyzed in conjunction with TCR sequences.

Main findings are:

- The T-cell repertoire is much smaller in AML patients compared to healthy bone marrow samples.
- a relative expansion of unconventional T cell types after treatment
- Presence of T cells from the same clonotype in several different maturation/activation subsets as defined by gene expression profiles.
- A relatively prominent GZMK+ T-cell population in AML patient which presumably reflects an intermediate differentiation stage.
- several findings from comparisons of serial samples are related to the treatment response, e.g. relative contraction versus expansion of T-cell clones; a weakly significant higher fraction of GZMK+ CD8 cells in responders; highly significant lower exhaustion scores in T cells of responders
- del11/q7 as a possible negative predictor for treatment response

The data and their analysis appear technically sound, but are mostly descriptive and exploratory. Data interpretation is hampered by the lack of appropriate control analyses, e.g. patients treated with azacytidine only. Few statistically significant findings are reported. In many instances where novel observations appear to be made, their validity cannot be ascertained by analysis of a validation cohort.

In line with the descriptive character of the manuscript, no specific hypotheses are being formulated, and many findings are described in comparison to immune responses to checkpoint inhibition in solid tumors. Since the mutanome in AML is much smaller than in tumors responding to checkpoint inhibition, and since the cited graft-versus-leukemia effect after allogeneic stem cell transplantation is biologically predominated by a graft-versus-hematopoiesis effect directed against minor histocompatibility antigens expressed in hematopoietic cells including AML, it is speculative in how far findings in solid tumors can be extrapolated and applied to AML.

Specific criticisms:

- 1) Fig 2A is unintelligible: What does a single data point on the graph indicate?
- 2) Fig 2C: Time point C is mentioned in the text but not displayed.
- 3) The scientific style is frequently misleading. E.g., the term significant should be used exclusively in the context of an appropriate statistical context.
- 4) There numerous typographical mistakes throughout the manuscript.
- 5) The sentence "The clonotype size in healthy BMs ranged from 1 to 16 TCR clonotypes, compared to 1 to 1200 TCR clonotypes in AML" does not make sense semantically

REVIEWER COMMENTS

Reviewer #1 (Remarks to the Author):

Abbas et al. use a state-of-the-art combination approach of single cell gene expression profiling and single cell TCR sequencing to assess the T cell phenotype and clonotype in BM cells from AML patients treated with anti-PD-1 (nivolumab) plus azacitidine combination therapy at a single cell level. BM samples were derived from 8 patients enrolled in the NCT02397720 study of nivolumab+azacitidine in r/r AML and 2 healthy donors. Paired scRNA and scTCR profiling revealed changes in cellular subset composition and TCR clonotypes pre- and post nivolumab+azacitidine treatment. They found that the most abundant clonotypes expanded primarily in patients that responded or had stable disease while the most abundant clonotypes retracted in non-responders. Furthermore, the authors suggest GZMK expression in CD8 T cells and MAIT cells as a potential marker for improved outcome in AML. Finally, they identified that chromosome 7/7q loss was associated with resistance to anti-PD-1/azacitidine treatment in a larger cohort of 57 patients. While this study provides an in-depth look at T cell phenotypes and TCR repertoires in patients before and after anti-PD-1 based therapy, one major concern is that it remains at a merely descriptive level and I am not not convinced that it will be practice changing in the clinic. In addition, a major part of the results is based on quantitative assessment of cell population proportions, which has to be viewed with caution because of the limited number of patients assessed. While differences in gene signatures can be quantitatively assessed using scRNA-seq, mere quantification of cell populations should preferably be analyzed with other methods that allow for larger patient and cell numbers, e.g. flow cytometry.

Major Comments

- The authors should be cautious with claiming quantitative changes in certain cell types (e.g. CD4 vs. CD8 cell subset) due to the very low patient numbers per group. It cannot be ruled out that single patients with high cell numbers drive specific changes. Single cell sequencing should be rather utilized to compare changes in gene expression in specific cell subsets between the patient groups, e.g. did T cell effector signatures change in the CD8 subset in response to treatment. Quantitative changes in cell subset proportions should rather be confirmed by flow with a) larger patient numbers and b) larger cell numbers, to be able to statistically assess these changes.

Response: We would like to thank the reviewer for this comment. We agree that single cell sequencing has its shortcomings in quantifying cellular distributions, although it has been used to identify cellular subsets via canonical marker characterization as previously described⁵⁻¹¹. We also correlated the percentage of blast cells detected with single cell RNA to that detected by flow cytometry and histologic review by experienced hematopathologists on the same bone marrow aspirate/biopsy and found very high concordance of blast frequencies, as previously undertaken⁴. To further quantify some of the findings in our analysis due to the limited tissue availability, we leveraged existing baseline flow cytometry data from 33 (13 responders and 20 non-responders) patients done as part of a correlative immune profiling at enrollment in patients enrolled to the azacitidine with nivolumab clinical trial. Prior to treatment initiation, we found significantly higher CD3+ (p=0.027) and CD3+ CD8+ (p=0.044) cells in responders. We have added this data in **Supplementary Figures 4A-B**. Furthermore, **Supplementary Figures 4C-H** list the cellular subsets by patients and timepoint, in addition to plotting the error bars, which demonstrate that the findings were not driven by 1 patient in our single cell analysis, except for MAIT cells which were indeed primarily contributed by 1 patient. To that end we highlighted this in the text and

excluded the MAIT cells from the CD4+ and CD8+ subset analysis. For patient-by-patient contribution to each cellular subset, we added **Supplementary Figure 4D**.

To note, our approach of single cell sequencing of all the bone marrow cells rather than sorted or select cellular subsets was undertaken to ensure a representative reflection of the heterogeneity of the cellular components in the bone marrows of different AML patients which is further supported by the large number of cells sequenced per patient and the depth of the sequencing as noted by high number and reads of transcripts. Specifically, we sequenced 113,394 cells from AML patients, which is higher than previous single cell work in AML²⁻⁴. We also covered on average 5,336 [range: 976-8,307] cells per sample with a median of 2,179 [range: 201-8,972] transcripts per cell. These parameters allow us to uncover more patient-specific cellular subsets.

We agree with the reviewer that the focus of the single cell profiling was indeed to identify differentially expressed genes within cellular subsets. To that end, we focused on CD8 GZMK cells and looked at the differential gene expression between responders and non-responders in CD8 GZMK cells (**Supplementary Figure 4G and Supplementary Table 2**). We identified 137 significantly upregulated and 455 significantly downregulated genes ($fdr < 0.05$) in the CD8 GZMK cells of responders compared to non-responders. Interestingly, pathway enrichment of upregulated genes confirmed significant enrichment of TNF-alpha signaling genes, which has been shown to be a significant feature of activate memory T cell activation^{20,21}. We have now added this data to the updated submitted manuscript.

- Why is timepoint C not presented in Figure 2C? Since this is a merely descriptive depiction of the clonality, all timepoints should be considered and reported. Does the time point "post" in Figure 2B and 3B refers to time point B or C – in particular in responders?

Response: Based on the reviewer's recommendations, we have now added timepoint C to **Figure 2C**. Please note that 2 of the patients (PT4 and PT6) did not have a timepoint C collection for the TCR data, which is why it was not included in the initial submission.

- Did decrease in clonality at timepoint C predict subsequent relapse? Did patients with persistent increase in T cell clonality stay in remission longer?

Response: Interestingly, PT3C (the third timepoint prior to relapse in patient 3) had decreased clonality and this heralded the relapse and the acquisition/emergence of chromosome 7 deletion (**Figure 6C**). PT1 who had the longest response, exceeding 3.5 years on the trial, and PT2 who was still responding at timepoint C, both had persistently higher TCR clonality at timepoint C.

- In Figure 3G the text implies that only responders are depicted. How did the exhaustion scores change in non-responders?

Response: Based on the recommendation of the reviewer, we now measured the exhaustion scores of non-responders. Consistent with the lower fraction of CTL in non-responders following treatment, exhaustion scores of CD8+ cells significantly decreased ($p=0.0005$). We added this new data and the percentages of T cells subtypes pre/post-treatment in **Supplementary Figures 4A and 5A**.

- Pseudotime analysis revealed that GZMB expression was higher at a later pseudotime. Did GZMB expression correlate with exhaustion signature?

Response: Per the reviewer's suggestion, we calculated the correlation between GZMB and the calculated exhaustion scores in CD8 cells. There was a weakly positive correlation ($r=0.1$; $p<0.01$). Therefore, while GZMB is expressed in CD8 Cytotoxic T lymphocyte at later pseudotime, the correlation with the exhaustion score was small. Data shown here:

- The findings of co-expression of TCF7 and EOMES in the GZMK+ CD8 T cells, as well as their stem cell like properties are interesting, but rather descriptive. The manuscript would be improved by confirming these findings by flow and in functional assays.

Response: We agree with the reviewer that it would be interesting to validate some of these findings pertaining to the memory signature of T cells. We used established canonical markers^{22,23} to maintain consistency in defining cellular subsets similar to what other groups have done in single cell profiling of cellular subsets from patient samples and used previously reported markers (TCF7 and EOMES)²⁴⁻²⁶ of stem-like and memory cells. Interestingly, in our analysis, these were enriched in the CD8 GZMK population which suggested its memory phenotype. In support of our findings:

1) While preparing this manuscript and since its submission, a study by Galletti et al¹² demonstrated that GZMK is indeed enriched in different subsets of memory T cells and crucial for their function.

2) Another study in colorectal cancer identified GZMK in a subset of memory cells using single cell profiling¹³

3) Using INs-seq that leverages single cell RNA and intracellular protein levels, Katzenelengbogen et al (2020)²⁷ demonstrated that flow-sorted human T cells can be distinguished by GZMK for 2 subsets of memory cells.

4) To further validate our findings in independent datasets, we now interrogated the Human Protein Atlas²⁸ data and evaluated the expression of GZMK cells in **1)** human blood cells ²⁹ **2)** Monaco Dataset ³⁰ **3)** Immune cell gene expression dataset blood cell types ³¹. In all 3 resources, the highest expression is seen in memory T cells providing an independent and orthogonal mode to support our data. We updated the results sections and added the data in **Supplementary Figure 5E-G**.

5) Finally, based on the human protein atlas database, lymphoid tissues, especially tonsils, are enriched for GZMK. We therefore conducted co-immunofluorescence on human tonsil tissue and verified the co-expression of the memory marker CD45RO with CD8 and GZMK, confirming that GZMK is indeed expressed in subset of memory cells. We updated the results sections and added the data in **Supplementary Figure 5H-K**.

- The MAIT cell data are interesting but have to be viewed with caution, since 90% of the cells were contributed by a single patient.

Response: We completely agree with the reviewer. We have indicated in the manuscript that conclusions pertaining to MAIT cells are primarily specific to one patient (PT1). However, the interesting clinical course of this patient who is alive to this date, 9 years after diagnosis with AML and now 94 years old was intriguing, even if an n of only 1. We think that MAIT cells may indeed be contributing to his leukemic control given that these cells significantly expanded following treatment with azacitidine with nivolumab in this patient, further characterizing patient-specific characteristics and emergent cellular populations that would otherwise have been missed in bulk analysis.

- The authors identify chr7/7q deletion as a resistance mechanism to checkpoint blockade and attribute this to expansion of regulatory T cells and reduced IFN γ signaling. What were the percentages/phenotype of Tregs in the patients in this study at different time points? Were these correlated with the duration of response/survival? Also here, functional assays to explore the role of IFN γ signaling in response to azacitidine/nivolumab would be highly beneficial.

Response: We evaluated whether T_{reg} frequency was correlated with outcomes and found no significant correlation between baseline Tregs and overall survival ($r=0.02$; $p=0.89$). The added **Supplementary Figure 4D and 4H** exhibits the contribution of each T cell subset, including T_{reg}, by each patient. The fraction of T_{reg} cells out of total CD4 cells is here.

In **Supplementary Figure 4H**, we show no significant difference in T_{reg} population prior to treatment across response groups.

We agree that our data pertaining to IFN γ requires substantial functional support to make a definitive conclusion. However, in our manuscript we merely investigated whether there were any genes on chromosome 7/7q region that potentially associated with the mechanisms of resistance and in an independent analysis IFN γ pathway was identified. However, we don't think our data is sufficient to support that this is the main pathway that is modulated and driving resistance.

Based on another reviewer's recommendation, we reanalyzed the data on chromosome 7/7q by loosening the inclusion criteria to include any differentially expressed genes with a false discovery rate (q value) <0.05. We evaluated differentially expressed genes between patients with intact chromosome 7 to patients with loss in chr7/7q. Of the 6400 upregulated and 2816 downregulated genes, 58 (0.9%) and 239 (8.5%) were on chr7q (Fisher's Exact test, p = 5.255048-73), listed now in **Supplementary Table 3**. As expected, chr7q deletion has resulted in significant transcriptome downregulation of chr7q genes consistent with gene dose effect. Pathway analysis of all differentially expressed genes on chr7q region revealed enrichment of genes from mTORC1 signaling, glycolysis and interferon-gamma and is now added as **Supplementary Figure 7A**. Among the genes on chromosome 7q that are also detected in our single cell RNA profiling, the IFN γ pathway genes were also significantly enriched. We modified our wording in the result and discussion sections to clarify the analysis.

Minor Comments

- Figure 2B and 3B would be clearer if the response of the individual patients was indicated as in Figure 6A.

Response: We updated Figures 2B and 3B to reflect color-coding for responses to make it clearer for readers.

- Please carefully check for typos:

o p.7 "... in clinical trial settings"

o p.2 "... stem-like properties likely." This sentence ends openly.

o p.10 "...expression levels of the cytotoxic gene GNLY, which delivers..."

Response: We apologize for these typos. We reviewed the manuscript and rectified any typographical and grammatical errors.

Reviewer #2 (Remarks to the Author):

In this manuscript, the authors performed single cell RNA-sequencing on a large number of bone marrow cells from patients with relapsed/refractory AML both pre- and post-treatment alongside healthy donors. They also profiled T-cell receptors in those same cells. In summary, the authors discovered drug response and cell type proportion heterogeneity within the T-cells, along with variable TCR repertoires between responders and non-responders. Their analysis found a developmental continuum of T-cells with differing levels of the GZMK gene as well as a potential mechanism for resistance to PD-1 blockade therapy. Overall, the authors effectively use biological and computational techniques in this manuscript to evaluate their hypothesis. The manuscript is clearly written, and the results novel and of value to the scientific community.

1. The author's state: "We identified 5 (2 conventional and 3 unconventional)48 T cell phenotypes in 25,798 T cells from 22 BM aspirates before and after treatment in AML patients (Fig. 3A). The 2 conventional phenotypes were CD4+ and CD8+ cells, constituting 53% and 35% of BM T cells at pretreatment, and 30.9% and 37.4% of BM T cells at posttreatment, respectively." This section goes further to highlight cell type proportion differences between highly similar cell types, such as CD8 and MAIT cells. I do not believe that bioinformatics tools in their current state are accurate enough in cell type identification to allow for definitively stating cell type proportion differences that are only moderately altered between the two patient groups. This section could be further supported using experimental methods to verify these proportion differences in some samples.

Response: We thank the reviewer for the overall positive comments regarding the manuscript and agree that single cell technologies, especially when applied in a small number of patients, have limitations in delineating certain cellular subsets. To note, our approach of single cell sequencing of all the bone marrow cells rather than sorted or select cellular subsets was undertaken to ensure a more representative reflection of the heterogeneity of the cellular components in the bone marrows of different AML patients which is further supported by the large number of cells sequenced per patient and the high depth of the sequencing. Specifically, we sequenced 113,394 cells from AML patients, which is higher than previous single cell work in AML²⁻⁴. We also covered on average 5,336 (range: 976-8,307) cells per sample with a median of 2,179 (range: 201-8,972) transcripts per cell. These parameters allow us to uncover more patient-specific cellular subsets and were concordant with a heterogenous tumor microenvironment, similar to other studies in solid cancer^{5-11,23,32-37}.

For cellular identity, we used canonical markers as previously done^{22,23}. In regards to MAIT cells, we used the conserved amino acid sequence derived from scTCR sequencing in addition to the TRAV1-2 marker to define these cells. Since flow cytometry data is available for blast cells, we compared blast percentage from flow cytometry as well as from the immunohistochemistry with the single-cell data and found a strong correlation between the percentage of blast cells detected in our single-cell analysis compared to the same bone marrow aspirate/biopsy timepoint analyzed via flow cytometry and histologic profiling. The histologic profiling and flow cytometry were reviewed by expert hematopathologists at our center, and as an additional layer for consistency we also inferred the cytogenetic changes by scRNA and compared it to the available cytogenetic data by routine karyotyping at time of sample collection and found them to be concordant. To further highlight to readers the distribution of the different tumor microenvironment cellular subsets and the heterogeneity across individual AML patients, we have now added **Supplementary Figure 4D** wherein we calculated the fraction of the different T cell subsets contributed by each

patient. **Figure 1I** in the main text also illustrates the distribution of all cells in the microenvironment of each timepoint.

2. In a couple of places in the manuscript the authors acknowledge the heterogeneity in the patient population. One case is the following: “Of note, 89.9% of MAIT cells in our analysis were contributed by PT1 (responder) who had a unique clinical course.” Given such a dramatic patient-specific effect, how are the authors assessing the significance of their results?

Response: We agree with the reviewer that the course of PT1 was quite dramatic and significantly different from other patients in the analysis, which is one of the reasons we highlighted this patient in the text. PT1 is the only patient who is still alive today at age of 94 years, 9 years since his diagnosis with AML (AML diagnosed at 85 years of age). He was the patient with the longest time on the azacitidine with nivolumab trial (3.5 years). Given that MAIT cells were highly enriched in the bone marrow and TCR expansions in PT1 occurred in the MAIT cell population following PD-1 blockade therapy, we thought it would be relevant to highlight his clinical course and these findings, suggesting some form of unique sensitivity and T-cell modulation with azacitidine with nivolumab in this patient. MAIT cells are emerging as potential targets for anticancer therapies based on their invariant recognition of MR1 molecules^{38,39}. We highlighted in the manuscript that these findings are pertinent just to one patient, however. To be as comprehensive as possible, the patient composition for T-cell subtypes by patient is now added (**Supplementary Figure 4D**). Noteworthy, for CD4 and CD8 analysis, there has been significant contribution of all patients to these cellular subsets.

3. In the Chr7/7q section, the authors found 13 significantly downregulated genes on chr7q in AML cells and used these genes to perform gene set enrichment analysis. This section could be improved by relaxing the significance criteria and performing the analysis with a larger set of genes in addition to the original analysis, as 13 is a very small gene set.

Response: We reanalyzed the data on chromosome 7/7q by loosening the inclusion criteria to include any differentially expressed genes with a false discovery rate (q value) <0.05. We evaluated differentially expressed genes between patients with intact versus loss in chr7/7q. Of the 6400 upregulated and 2816 downregulated genes, 58 (0.9%) and 239 (8.5%) were on chr7q (Fisher’s Exact test, $p = 5.255048 \times 10^{-73}$), listed now in **Supplementary Table 3**. As expected, chr7q deletion has resulted in significant transcriptome downregulation of chr7q genes consistent with gene dose effect. Pathway analysis of all differentially expressed genes on chr7q region revealed enrichment of genes from mTORC1 signaling, glycolysis and interferon-gamma and is now added as **Supplementary Figure 7A**. Among the genes on chromosome 7q that are also detected in our single cell RNA profiling, the IFN γ pathway genes were also significantly enriched.

4. The author’s state that “All datasets generated during and/or analyzed during the current study will be available from the corresponding authors.” Will these data become publicly available upon publication?

Response: scRNA and scTCR for all 22 AML timepoints and the 2 healthy bone marrows are now submitted to EGA. As soon as we receive the EGA access code, we will send it to the editorial office. The data will also be made public once the manuscript is accepted in accordance with Nature Communications publishing policy.

5. Doublets were identified using robust criteria, which should remove doublets effectively, but new computational tools for doublet detection, such as Scrublet, DoubletDecon, and

Solo, have shown very high accuracy in biologically validated datasets. Given the high transcriptional similarity between the T-cell populations of interest and the difficulty in identifying doublets because of this with the given techniques, I would suggest using a computational approach for further validation.

Response: Scrublet¹⁷ and DoubletFinder¹⁸ use similar strategies and outperform other algorithms in identifying doublets. Based on a recent benchmarking study¹⁹, DoubletFinder achieves the best detection accuracy for doublet cells. We thus performed the doublet identification using DoubletFinder (v2.0.3) on our cells that passed our initial QC. Since DoubletFinder always identifies a subset of cells as doublet based on the user-defined parameter “nExp”, we used the doublet score rather than the predicted doublets to test whether we can observe doublet clusters enriched with high doublet score. According to the doublet score distribution that we now generated (**Supplementary Figure 2F-G**), we didn’t observe such clusters, suggesting that doublets have been removed by and large. We have added and updated the manuscript text, supplementary figures and also updated the methods section to reflect these additional QC steps.

6. The legend for Figure 1C refers to a UMAP-based pseudotemporal trajectory analysis, but figure shows only the UMAP and cell type proportion predictions with no trajectory analysis.

Response: The reviewer is correct and thanks for catching this. We fixed the legend to reflect the UMAP trajectory, while we now added the pseudotemporal trajectory as **Supplementary Figure 2A**.

7. In Figure 1D, my immediate concern with the green and red UMAP was potential batch effects, given the separation between the PT1C and Normal BM cells in non-malignant cell types. Upon further reflection, this could simply be an overplotting issue. Using side-by-side UMAPs in the same dimensionality reduction space would better highlight the integration and differences and alleviate batch effect concerns.

Response: We agree with the reviewer. The batch effect has been corrected using Harmony⁴⁰, and it was an overplotting issue. Per the reviewer’s recommendation, we replotted the NL and AML of PT1C separately and updated Figure 1D.

8. For Figure 3A, within each cell type do the distinct subclusters align with the mutation breakdown? If not, what transcriptional change causes the small subsets of cells to distinctly separate from the larger cell population?

Response: We apologize if the text was not clear. In Figure 3A, we are projecting T-cells only. Our approach doesn’t allow us to identify mutations in T cells from our scRNA sequencing. The small cluster that the reviewer is referring to is actually Th17 which we verified using marker genes as listed in Figure 3C and further supported by canonical gene markers as previously described^{22,23}.

Reviewer #3 (Remarks to the Author):

The authors analyze single-cell transcriptomics data from serial bone marrow samples of 8 patients treated for relapsed/refractory AML with azacytidine and the checkpoint inhibitor nivolumab. With respect to immune cells, transcriptomic data are analyzed in conjunction with TCR sequences.

Main findings are:

- *The T-cell repertoire is much smaller in AML patients compared to healthy bone marrow samples.*
- *a relative expansion of unconventional T cell types after treatment*
- *Presence of T cells from the same clonotype in several different maturation/activation subsets as defined by gene expression profiles.*
- *A relatively prominent GZMK+ T-cell population in AML patient which presumably reflects an intermediate differentiation stage.*
- *several findings from comparisons of serial samples are related to the treatment response, e.g. relative contraction versus expansion of T-cell clones; a weakly significant higher fraction of GZMK+ CD8 cells in responders; highly significant lower exhaustion scores in T cells of responders*
- *del1/q7 as a possible negative predictor for treatment response*

The data and their analysis appear technically sound, but are mostly descriptive and exploratory. Data interpretation is hampered by the lack of appropriate control analyses, e.g. patients treated with azacytidine only. Few statistically significant findings are reported. In many instances where novel observations appear to be made, their validity cannot be ascertained by analysis of a validation cohort.

In line with the descriptive character of the manuscript, no specific hypotheses are being formulated, and many findings are described in comparison to immune responses to checkpoint inhibition in solid tumors. Since the mutanome in AML is much smaller than in tumors responding to checkpoint inhibition, and since the cited graft-versus-leukemia effect after allogeneic stem cell transplantation is biologically predominated by a graft-versus-hematopoiesis effect directed against minor histocompatibility antigens expressed in hematopoietic cells including AML, it is speculative in how far findings in solid tumors can be extrapolated and applied to AML.

Response: We would like to thank the reviewer for the feedback and comments. We agree that several findings are descriptive in our manuscript, which is not uncommon in single cell studies especially derived from patient samples on clinical trials. However, single cell RNA sequencing could still be a powerful tool to understand tissue heterogeneity. We empowered our analysis by conducting shared RNA and TCR analysis which to our knowledge is the first demonstration in AML to identify the expansions, contractions and emergence of novel clonotypes in response to PD-1 blockade-based therapy. Further, our longitudinal analysis allowed us to understand the dynamics of response in some patients. For instance, we found increased TCR clonality in responders. Interestingly, TCR clonality in PT1 and PT2 (both responders), was high at the third timepoint while response was still notable. However, in PT3 (responder), TCR clonality decreased at timepoint C which preceded the chromosome 7/7q acquired loss and relapse of AML. This demonstrated that T-cell activities are likely responsible for the response in therapy. Further, from PT1, we learnt about MAIT cell expansion in PT1 who had >3.5 years of response while on this therapy and is alive to date at age of 94 years, 9 years after AML diagnosis. We were also able to characterize the different CD8 subsets in AML bone marrow using multiple gene markers and provided a pseudotemporal analysis of the granzyme-based subsets in bone marrows, which we further verified now in our updated analysis in 3 independent datasets. Finally, we agree that the mutanome in AML is much smaller. Therefore, we investigated large cytogenetic changes which are more common in AM. Our findings support that the large cytogenetic losses, specifically chromosome 7/7q loss, was associated with resistance to PD-1 blockade therapy.

Regarding a control subset, we analyzed a larger cohort of patients treated with hypomethylating agents-based therapy but without immunotherapy and found no correlation with chromosome 7/7q for resistance in the relapsed/refractory setting. Of note, 7 out of the 8 patients had received hypomethylating agents (azacitidine or decitabine) prior to enrollment and all had been either non-responders or relapsed while on that treatment. We therefore think that the findings we are seeing in our data are largely driven by PD-1 blockade.

We agree with the reviewer that the application of these findings may be distinct from what is seen in solid cancers, and we think that is a major highlight since learning from solid cancers in immunotherapy may not apply in AML and thus warrant further investigation. To further improve our analysis and support our findings, we would like to note the following:

- 1) While preparing this manuscript and since its submission, a study by Galletti et al¹² demonstrated that GZMK is indeed enriched in different subsets of memory T cells and crucial for their function.
- 2) Another study in colorectal cancer identified GZMK in a subset of memory cells using single cell profiling¹³
- 3) Using INs-seq that leverages single cell RNA and intracellular protein levels, Katzenelengbogen et al (2020)²⁷ demonstrated that flow-sorted human memory T cells can be distinguished by GZMK for 2 subsets of memory cells.
- 4) To further validate our findings in independent datasets, we now interrogated the Human Protein Atlas²⁸ data and evaluated the expression of GZMK cells in **1) human blood cells**²⁹ **2) Monaco Dataset**³⁰ **3) Immune cell gene expression dataset blood cell types**³¹. In all 3 resources, the highest expression is seen in memory T-cells providing an independent and orthogonal mode to support our data. We updated the results sections and added the data in **Supplementary Figure 5E-G**.
- 5) Finally, based on the human protein atlas database, lymphoid tissues, especially tonsils, are enriched for GZMK. We therefore conducted co-immunofluorescence on human tonsil tissue and verified the co-expression of the memory marker CD45RO with CD8 and GZMK, confirming that GZMK is indeed expressed in subset of memory cells. We updated the results sections and added the data in **Supplementary Figure 5H-K**.

We hope that our efforts will highlight the feasibility and depth of data obtainable from single cell RNAseq on clinical trial samples, thereby encouraging other investigators and sponsors to consider this approach on future prospective immune or targeted clinical trials in patients with AML.

Specific criticisms:

1) *Fig 2A is unintelligible: What does a single data point on the graph indicate?*

Response: We apologize if this was unclear. We had used the same analysis approach as previously reported by Guo et al¹⁰. Briefly, the scatter plot demonstrates the correlation between the distinct number of T cell clonotypes and the size of the clonotype i.e. the number of T cells contributing to the clonotype. The dashed line separates singletons (non-clonal cells) and multiplsets (i.e. clonal cells). The findings in the figure support clonal expansions in the tumors as represented by high number of T cells per clonotype, suggesting higher clonality. We further clarified this in the figure legend.

2) *Fig 2C: Time point C is mentioned in the text but not displayed.*

Response: We have now updated the Figure to include all timepoints. Interestingly, PT3C (the third timepoint prior to relapse in PT3) had decreased clonality and this heralded the relapse and the acquisition/emergence of chromosome 7 deletion (**Figure 6C**). PT1 who had the longest response, exceeding 3.5 years on the trial, and PT2 who was still responding at time of sequencing, both had persistently higher TCR clonality at timepoint C.

3) *The scientific style is frequently misleading. E.g., the term significant should be used exclusively in the context of an appropriate statistical context.*

Response: We apologize for inappropriately using “significantly” in certain instances. We have reviewed the manuscript and removed statements of ‘significance’ that were rather descriptive and not statistical. We also added statistical tests and measures for statements with statistical significance throughout all the manuscript.

4) *There numerous typographical mistakes throughout the manuscript.*

Response: We reviewed the manuscript and rectified any grammatical and typographical errors.

5) *The sentence "The clonotype size in healthy BMs ranged from 1 to 16 TCR clonotypes, compared to 1 to 1200 TCR clonotypes in AML" does not make sense semantically*

Response: We changed the sentence. It now reads “The clonotype size, represented by number of cells expressing the same TCR sequence, ranged from 1 to 16 in healthy donors, compared to 1 to 1200 in AML, indicating higher clonality in AML bone marrows.”

Rebuttal References

1. Daver, N. *et al.* Efficacy, Safety, and Biomarkers of Response to Azacitidine and Nivolumab in Relapsed/Refractory Acute Myeloid Leukemia: A Nonrandomized, Open-Label, Phase II Study. *Cancer Discov* **9**, 370–383 (2019).
2. Sachs, K. *et al.* Single-Cell Gene Expression Analyses Reveal Distinct Self-Renewing and Proliferating Subsets in the Leukemia Stem Cell Compartment in Acute Myeloid Leukemia. *Cancer Res.* **80**, 458–470 (2020).

3. Thomas, B. E., Perumalla, P., Bhasin, S. S. 2020. *Single Cell Transcriptomics Revealed AML and Non-AML Cell Clusters Relevant to Relapse and Remission in Pediatric AML.* (2021).
4. van Galen, P. *et al.* Single-Cell RNA-Seq Reveals AML Hierarchies Relevant to Disease Progression and Immunity. *Cell* **176**, 1265–1281.e24 (2019).
5. Savas, P. *et al.* Single-cell profiling of breast cancer T cells reveals a tissue-resident memory subset associated with improved prognosis. *Nature Medicine* **24**, 986–993 (2018).
6. Qian, J. *et al.* A pan-cancer blueprint of the heterogeneous tumor microenvironment revealed by single-cell profiling. *Cell Research* **30**, 1–18 (2020).
7. Chen, Y.-P. *et al.* Single-cell transcriptomics reveals regulators underlying immune cell diversity and immune subtypes associated with prognosis in nasopharyngeal carcinoma. *Cell Research* **30**, 1–19 (2020).
8. Gide, T. N. *et al.* Distinct Immune Cell Populations Define Response to Anti-PD-1 Monotherapy and Anti-PD-1/Anti-CTLA-4 Combined Therapy. *Cancer Cell* **35**, 238–255.e6 (2019).
9. Puram, S. V. *et al.* Single-Cell Transcriptomic Analysis of Primary and Metastatic Tumor Ecosystems in Head and Neck Cancer. *Cell* **171**, 1611–1624.e24 (2017).
10. Guo, X. *et al.* Global characterization of T cells in non-small-cell lung cancer by single-cell sequencing. *Nature Medicine* **24**, 1–17 (2018).
11. Gubin, M. M. *et al.* High-Dimensional Analysis Delineates Myeloid and Lymphoid Compartment Remodeling during Successful Immune-Checkpoint Cancer Therapy. *Cell* **175**, 1014–1030.e19 (2018).
12. Galletti, G. *et al.* Two subsets of stem-like CD8+ memory T cell progenitors with distinct fate commitments in humans. *Nat Immunol* **21**, 1–20 (2020).
13. Zhang, L. *et al.* Lineage tracking reveals dynamic relationships of T cells in colorectal cancer. *Nature* 1–30 (2018). doi:10.1038/s41586-018-0694-x
14. Katzenelenbogen, Y. *et al.* Coupled scRNA-Seq and Intracellular Protein Activity Reveal an Immunosuppressive Role of TREM2 in Cancer. *Cell* **182**, 872–885.e19 (2020).
15. Lafzi, A., Moutinho, C., Picelli, S. & Heyn, H. Tutorial: guidelines for the experimental design of single-cell RNA sequencing studies. *Nat Protoc* **13**, 2742–2757 (2018).
16. Lähnemann, D. *et al.* Eleven grand challenges in single-cell data science. *Genome Biol.* **21**, 31–35 (2020).
17. Wolock, S. L., Lopez, R. & Klein, A. M. Scrublet: Computational Identification of Cell Doublets in Single-Cell Transcriptomic Data. *Cell Syst* **8**, 281–291.e9 (2019).
18. McGinnis, C. S., Murrow, L. M. & Gartner, Z. J. DoubletFinder: Doublet Detection in Single-Cell RNA Sequencing Data Using Artificial Nearest Neighbors. *Cell Syst* **8**, 329–337.e4 (2019).
19. Xi, N. M. & Li, J. J. Benchmarking Computational Doublet-Detection Methods for Single-Cell RNA Sequencing Data. *Cell Syst* **12**, 176–194.e6 (2021).
20. Scott, A. C. *et al.* TOX is a critical regulator of tumour-specific T cell differentiation. *Nature* **571**, 270–274 (2019).
21. Ye, L.-L., Wei, X.-S., Zhang, M., Niu, Y.-R. & Zhou, Q. The Significance of Tumor Necrosis Factor Receptor Type II in CD8+ Regulatory T Cells and CD8+ Effector T Cells. *Front Immunol* **9**, 583 (2018).
22. Lambrechts, D. *et al.* Phenotype molding of stromal cells in the lung tumor microenvironment. *Nature Medicine* **24**, 1277–1289 (2018).
23. Sade-Feldman, M. *et al.* Defining T Cell States Associated with Response to Checkpoint Immunotherapy in Melanoma. *Cell* **175**, 998–1013.e20 (2018).
24. Yost, K. E. *et al.* Clonal replacement of tumor-specific T cells following PD-1 blockade. *Nature Medicine* **25**, 1251–1259 (2019).

25. Knox, J. J., Cosma, G. L., Betts, M. R. & McLane, L. M. Characterization of T-bet and eomes in peripheral human immune cells. *Front Immunol* **5**, 217 (2014).
26. Istaces, N. *et al.* EOMES interacts with RUNX3 and BRG1 to promote innate memory cell formation through epigenetic reprogramming. *Nature Communications* **10**, 3306–17 (2019).
27. Katzenelenbogen, Y. *et al.* Coupled scRNA-Seq and Intracellular Protein Activity Reveal an Immunosuppressive Role of TREM2 in Cancer. *Cell* **182**, 872–885.e19 (2020).
28. Uhlen, M. *et al.* Proteomics. Tissue-based map of the human proteome. *Science* **347**, 1260419 (2015).
29. Uhlen, M. *et al.* A genome-wide transcriptomic analysis of protein-coding genes in human blood cells. *Science* **366**, (2019).
30. Monaco, G. *et al.* RNA-Seq Signatures Normalized by mRNA Abundance Allow Absolute Deconvolution of Human Immune Cell Types. *Cell Rep* **26**, 1627–1640.e7 (2019).
31. Schmiedel, B. J. *et al.* Impact of Genetic Polymorphisms on Human Immune Cell Gene Expression. *Cell* **175**, 1701–1715.e16 (2018).
32. Zhang, J.-Y. *et al.* Single-cell landscape of immunological responses in patients with COVID-19. *Nat Immunol* **21**, 1107–1118 (2020).
33. Maynard, A. *et al.* Therapy-Induced Evolution of Human Lung Cancer Revealed by Single-Cell RNA Sequencing. *Cell* **182**, 1232–1251.e22 (2020).
34. Izar, B. *et al.* A single-cell landscape of high-grade serous ovarian cancer. *Nature Medicine* **26**, 1271–1279 (2020).
35. Cheng, S. *et al.* A pan-cancer single-cell transcriptional atlas of tumor infiltrating myeloid cells. *Cell* **184**, 792–809.e23 (2021).
36. Zhang, L. *et al.* Single-Cell Analyses Inform Mechanisms of Myeloid-Targeted Therapies in Colon Cancer. *Cell* **181**, 442–459.e29 (2020).
37. Zheng, C. *et al.* Landscape of Infiltrating T Cells in Liver Cancer Revealed by Single-Cell Sequencing. *Cell* **169**, 1342–1356.e16 (2017).
38. Koay, H.-F. *et al.* Diverse MR1-restricted T cells in mice and humans. *Nature Communications* **10**, 2243–15 (2019).
39. Godfrey, D. I., Uldrich, A. P., McCluskey, J., Rossjohn, J. & Moody, D. B. The burgeoning family of unconventional T cells. *Nat Immunol* **16**, 1114–1123 (2015).
40. Korsunsky, I. *et al.* Fast, sensitive and accurate integration of single-cell data with Harmony. *Nature Methods* **16**, 1289–1296 (2019).

REVIEWERS' COMMENTS

Reviewer #1 (Remarks to the Author):

The authors have fully addressed my comments. The manuscript has improved during revisions.

Reviewer #2 (Remarks to the Author):

The authors have sufficiently responded to my concerns and updated the manuscript accordingly. They have also responded to the other reviewer concerns, judged to the best of my ability, leaving no further concerns with this manuscript.